# MULTIVARIATE TIME-SERIES IMPUTATION WITH DISENTANGLED TEMPORAL REPRESENTATIONS

**SHUAI LIU**
School of Computer Science and Engineering
Nanyang Technological University
50 Nanyang Avenue, Singapore, 639798
`SHUAI004@e.ntu.edu.sg`

**Xiucheng Li**✉
School of Computer Science and Technology
Harbin Institute of Technology (Shenzhen)
No.6, Pingshan 1st Road, Nanshan District,
Shenzhen, Guangdong, China, 518055
`lixiucheng@ hit.edu.cn`

**Gao Cong & Yile Chen & Yue Jiang**
School of Computer Science and Engineering
Nanyang Technological University
50 Nanyang Avenue, Singapore, 639798
`gaocong @ntu.edu.sg,{yile001,yue013}001@e.ntu.edu.sg`

## ABSTRACT

Multivariate time series often faces the problem of missing value. Many time series imputation methods have been developed in literature. However, they all rely on an entangled representation to model dynamics of time series, which may fail to fully exploit the multiple factors (e.g., periodic patterns) presented in the data. Moreover, the entangled representations usually have no semantic meaning, and thus they often lack interpretability. In addition, many recent models are proposed to deal with the whole time series to identify temporal dynamics, but they are not scalable to long time series. Different from existing approaches, we propose TIDER, a novel matrix factorization-based method with disentangled temporal representations that account for multiple factors, namely *trend*, *seasonality*, and *local bias*, to model complex dynamics. The learned disentanglement makes the imputation process more reliable and offers explainability for imputation results. Moreover, TIDER is scalable to long time series. Empirical results show that our method outperforms existing approaches on three typical real-world datasets, especially on long time series, reducing mean absolute error by up to 50%. It also scales well to long datasets on which existing deep learning based methods struggle. Disentanglement validation experiments further highlight the robustness and accuracy of our model.

## 1 INTRODUCTION

Multivariate time series analysis (e.g., forecasting (Zeng et al., 2021) and classification (Li et al., 2022)) has a wide spectrum of applications like traffic flow forecasting (Liu et al., 2020), electricity demand prediction (Kaur et al., 2021), motion detection (Laddha et al., 2021), health monitoring(Tonekaboni et al., 2021), etc. Most of these multivariate time series analysis approaches typically assume intact input for building models. However, real-world multivariate time series tends to have missing values caused by factors like device malfunction, communication failure, or costly measurement, leading to impaired performance of these approaches, or even rendering them inapplicable.

In light of this, many time series imputation methods have been proposed to infer missing values from the observed ones. A multivariate time series, denoted as $\mathbf{X} \in \mathbb{R}^{N \times T}$, consists of $N$ univariate time series (called channels) spanning over $T$ time steps. Hence, it offers two perspectives for imputation: modeling cross-channel correlations and exploiting temporal dynamics. Earlier methods (Batista et al., 2002; Acuna & Rodriguez, 2004; Box et al., 2015) either aggregate observed entries across channels by estimating similarity between distinct channels, or solely exploit local smoothness or linear assumption in the same channel to fill in missing values. Since these methods lack the ability of modeling nonlinear dynamics and complex correlations, they may not perform well in practice.

In recent years, deep learning based methods (Liu et al., 2019; Tashiro et al., 2021; Cini et al., 2022) were proposed for time series imputation. These methods typically employ Recurrent Neural Networks (RNNs) as the backbone to jointly model nonlinear dynamics along temporal dimension (by updating hidden states nonlinearly over time steps) and cross-channel correlations (by mapping $\mathbf{x}_t \in \mathbb{R}^N$ from data space to hidden state space). These RNN-based imputation models achieve better results by further combining with multi-task loss (Cao et al., 2018; Cini et al., 2022) or adversarial training (Liu et al., 2019; Miao et al., 2021). Despite their success, these methods rely solely on a single entangled representation (hidden state) to model dynamics. However, for many real-world multivariate time series, their dynamics are rich combinations of multiple independent factors like trend, seasonality (periodicity), and local idiosyncrasy (Woo et al., 2022). Modeling combinations of these factors with a single entangled representation may not give good performance, as the entangled representation has to compromise itself to explain multiple orthogonal patterns (like local changes or exogenous interventions vs global patterns) together (Bengio et al., 2013). Furthermore, it becomes exacerbated when seasonal patterns dominate, as RNNs lack the inductive bias to initiatively capture periodicity (Hewamalage et al., 2021). In addition, the hidden states that these models learned are entangled and complex combination of various components. Thus it is difficult for them to provide interpretable information to explain imputation. Moreover, these methods require the entire time series to be fed in to their models at each forward step, to capture temporal dynamics.This is prohibitively costly for large $T$. Therefore, these methods are not applicable to long datasets.

To address these limitations, in this paper, we propose a novel multivariate time series imputation method, Time-series Imputation with Disentangled tEmporal Representations (TIDER), in which we explicitly model the complex dynamics of multivariate time series with disentangled representations to account for different factors. We employ a low-rank matrix decomposition framework, and achieve the disentanglement by imposing different forms of constraints on different representations which compose the low-rank matrix. In particular, we introduce a neighboring-smoothness representation matrix to explain the trend, a Fourier series-based representation matrix to define periodic inductive bias, and a local bias representation matrix to capture idiosyncrasies specific to individual time step. These disentangled representations offer greater flexibility and robustness to model the factors contributed to the dynamics in time series. Another notable benefit of these semantically meaningful disentangled representations is that they offer TIDER an interpretable perspective on imputation. Moreover, TIDER is a scalable model. It can be applied to time-series datasets with large $T$ and achieve much better imputation performance. In summary, our contributions are as follows.

- We propose TIDER, a new multivariate time series imputation model, which is featured with effective and explainable disentangled representations to account for various factors that characterize the complex dynamics of time series. To the best of our knowledge, TIDER is the first model to learn disentangled representations for multivariate time series imputation.

- TIDER is the first imputation model that introduces a learnable Fourier series-based representation to capture periodic patterns inherent in time series data.

- Extensive experiments show that our proposed method outperforms the baseline imputation methods in terms of effectiveness and scalability. Especially, for imputation task in long time series, TIDER achieves more than 50% improvement in MAE, compared with the best imputation baseline approach. Furthermore, TIDER is a scalable model. It can easily handle long multivariate time series while existing deep-learning methods struggle. We also demonstrate the explanability of the disentangled representations with several case studies.

## 2 RELATED WORK

Early time series imputation methods based on simple statistical strategies focus on exploiting local smoothness of temporal dimension as well as the similarity between different channels. For example, SimpleMean/SimpleMedian (Fung, 2006) imputes missing values by averaging, and KNN (Batista et al., 2002) aggregates cross-channel observations to fill in missing slots with k-nearest neighbors. Linear dynamics-based imputation methods, including linear imputation and state-space models (Durbin & Koopman, 2012), have also been employed. MICE (Van Buuren & Groothuis-Oudshoorn, 2011) explores the idea of estimating missing slots with chain equations. These methods typically lack the ability of exploiting nonlinear dynamics and complex correlation across channels.

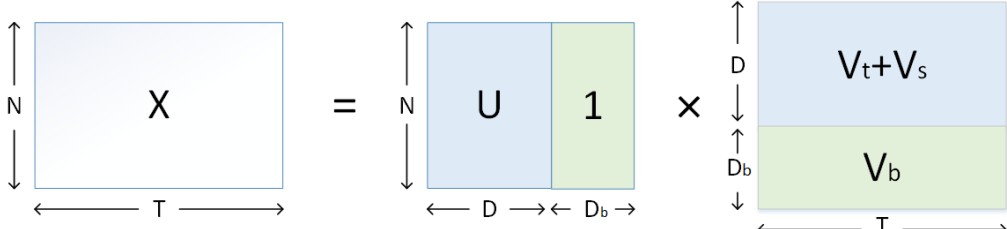

Figure 1: Architecture of the proposed method TIDER. $\mathbf{X}$ is the multivariate time series matrix. $\mathbf{U}$ represents a correlation-decoupled matrix. $\mathbf{V}_t, \mathbf{V}_s$, and $\mathbf{V}_b$ denote trend representation matrix, seasonality representation matrix and bias representation matrix, respectively.

A recent trend is to apply deep learning, particularly RNNs, to better capture the nonlinearity. BRITS (Cao et al., 2018) treats missing values as variables which participate in bidirectional RNN back-propagation and are updated with hidden states. GAIN (Yoon et al., 2018) combines Generative Adversarial Nets (GAN) and RNNs for imputation. NAOMI (Liu et al., 2019) merges RNNs and adversarial training strategies. SSGAN (Miao et al., 2021) is a semi-supervised GAN-based model which uses label information. GRIN (Cini et al., 2022) fuses graph message passing into GRU structure to learn spatial-temporal patterns. All these methods model dynamics based on a single entangled representation, which is insufficient to capture multiple factors underlying time series, especially when seasonality emerges, since RNNs lack the inductive bias to initiatively capture periodicity (Hewamalage et al., 2019). Moreover, these approaches are not scalable to long datasets since they have to process time series of whole length $T$ at each forward step to capture temporal dynamics. Thus, they do not perform well, or are even not applicable to long time-series data.

Our proposed TIDER is based on low-rank matrix factorization (MF) (Yu et al., 2016; Bjorck et al., 2021). Vanilla MF-based models impute missing entries by learning latent factors $\mathbf{U}, \mathbf{V}$ to exploit low-rank structure. However, they overlook the temporal continuity. To address this, TRMF (Yu et al., 2016) imposes autoregressive constraints on temporal factor $\mathbf{V}$. However, similar to aforementioned methods, TRMF also relies on entangled representation to account for all factors underlying the dynamics. In contrast, TIDER introduces multiple disentangled representations. It achieves the disentanglement by enforcing different forms of constraints into different representations.

## 3 PROBLEM STATEMENT

Given $N$ univariate time series data $\mathbf{x}_1, \mathbf{x}_2, \ldots, \mathbf{x}_N \in \mathbb{R}^T$ collected over $T$ time steps, we represent it with a multivariate time series matrix $\mathbf{X} \in \mathbb{R}^{N \times T}$, whose $n$-th row represents $n$-th univariate time series (channel) $\mathbf{x}_n$ and $t$-th column denotes the observation of all time series at time step $t$. The multivariate time series matrix $\mathbf{X}$ is incomplete and a fraction of entries are missed. We aim to infer missing values from the observed ones, and we denote the mask matrix as $\mathbf{M} \in \{0,1\}^{N \times T}$, where

$$M_{ij} = \begin{cases} 1, & \text{if } X_{ij} \text{ is observed,} \\ 0, & \text{otherwise.} \end{cases} \tag{1}$$

## 4 METHODOLOGY

### 4.1 METHOD OVERVIEW

The core idea of TIDER is to decompose multivariate time series $\mathbf{X}$ into two latent factors $\mathbf{U}$ and $\mathbf{V}$, such that $\mathbf{U}$ only preserves features unique to each channel whereas $\mathbf{V}$ is determined by multiple disentangled representations that jointly capture temporal dynamics. We employ such factorization for two main reasons: 1) univariate time series channels (rows of $\mathbf{X}$) are usually highly correlated; 2) observations at different time steps (columns of $\mathbf{X}$) exhibit strong temporal dynamics. The benefits of our design are twofold: 1) cross-channel correlations are decoupled since $\mathbf{U}$ only preserves channel-specific features 2) time-related information is isolated into $\mathbf{V}$, and it enables us to model the potentially complex temporal dynamics with multiple explainable disentangled representations.

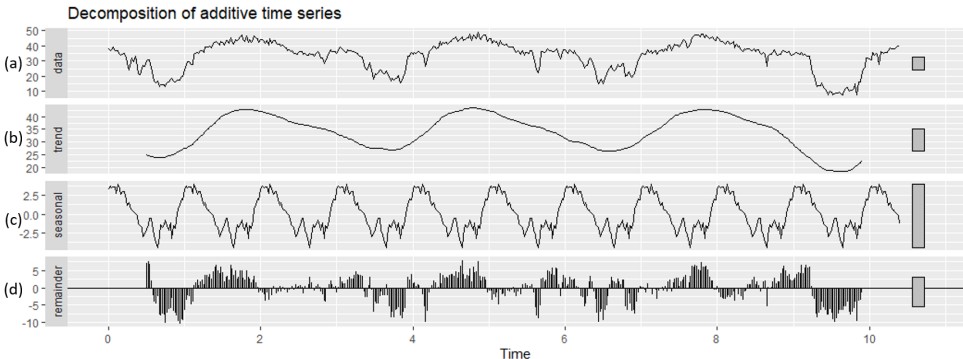

Figure 2: Visualization of time series decomposition. (a) is the raw time series, whereas (b), (c), and (d) are its decomposed trend, seasonality, and residual component. The grey bars on the right of each subplot show the relative scales of the components. Each grey bar represents the same length.

Figure 1 shows the architecture of TIDER. We adopt a low-rank matrix decomposition framework to factorize multivariate time series matrix $\mathbf{X}$ into two latent factors $\mathbf{U}, \mathbf{V}$. $\mathbf{U}$ is a correlation-decoupled matrix which accounts for channel-specific patterns, while $\mathbf{V}$ is a matrix for time-related information. Since temporal dynamics underlying real-world time series can be rich and complex combinations of multiple factors, e.g., trend and seasonality, modeling them merely through an entangled representation matrix will lead to model degradation. Woo et al. (2022) proposes that under mild assumptions, seasonality and trend can be treated as independent factors in time series generating process, and Cohen (2013) suggests that independence can be used as a proxy criterion for disentanglement. Inspired by them, we propose to model $\mathbf{V}$ with multiple disentangled representations, each accounting for one particular factor. We achieve the disentanglement by enforcing distinct forms of constraints on different representations, which introduce distinct inductive biases into these representations and make them more liable to capture specific semantically-independent patterns. More specifically, we consider three important factors: *trend*, *seasonality*, and *bias*, which are specified by three representation matrix $\mathbf{V}_t$, $\mathbf{V}_s$, $\mathbf{V}_b$, respectively. Figure 2 illustrates an example of decomposing a time series into these three factors. Trend representation matrix $\mathbf{V}_t \in \mathbb{R}^{D \times T}$ captures the intrinsic trend which changes gradually and smoothly, and seasonality representation matrix $\mathbf{V}_s \in \mathbb{R}^{D \times T}$ illustrates the periodic patterns hidden in temporal dynamics. $\mathbf{V}_t$ and $\mathbf{V}_s$ jointly determine the dynamics driven by endogenous factors. Bias representation matrix $\mathbf{V}_b \in \mathbb{R}^{D_b \times T}$ characterizes variations specific to each time step. Intuitively, $\mathbf{V}_b$ explains the individual idiosyncratic behaviors of time steps, which are orthogonal to global dynamics but shared across channels at a given time step. Hence, we treat it differently from $\mathbf{V}_t$, $\mathbf{V}_s$ and interpret it as a residual term matrix, i.e., $\mathbf{X} - \mathbf{U}(\mathbf{V}_t + \mathbf{V}_s) \approx \mathbf{1}_{N \times D_b} \mathbf{V}_b$. Mathematically, we formulate the objective of TIDER as

$$\text{minimize} \, \|(\mathbf{X} - \mathbf{U}_a \mathbf{V}) \odot \mathbf{M}\|^2 + \lambda_t f_t(\mathbf{V}_t) + \lambda_b f_b(\mathbf{V}_b) + \eta_1 \|\mathbf{U}\|^2 + \eta_2 \|\mathbf{V}\|^2,$$

$$\mathbf{U}_a = [\, \mathbf{U} \mid \mathbf{1} \,] \in \mathbb{R}^{N \times (D + D_b)}, \quad \mathbf{V} = \begin{bmatrix} \mathbf{V}_t + \mathbf{V}_s \\ \mathbf{V}_b \end{bmatrix} \in \mathbb{R}^{(D + D_b) \times T}, \tag{2}$$

where $\mathbf{U}_a$ is the augmented matrix of latent factor $\mathbf{U} \in \mathbb{R}^{N \times D}$, $f_t$ and $f_b$ are corresponding inductive bias constraint functions imposed on $\mathbf{V}_t$ and $\mathbf{V}_b$. $\mathbf{M}$ is the mask matrix introduced in Sec 3. $\eta_1 \|\mathbf{U}\|^2$, $\eta_2 \|\mathbf{V}\|^2$ are used to regularize the magnitude of latent factors, and $\lambda_t, \lambda_b, \eta_1, \eta_2$ are corresponding weights for each term. When training is completed, we use the learned $\mathbf{U}$ to get $\mathbf{U}_a$, and the learnt $\mathbf{V}_t$, $\mathbf{V}_s$, and $\mathbf{V}_b$ to form $\mathbf{V}$. $\mathbf{U}_a$ and $\mathbf{V}$ are then used to generate the imputed time series $\hat{\mathbf{X}}$ as

$$\hat{X}_{ij} = \begin{cases} X_{ij}, & M_{ij} = 1, \\ (\mathbf{U}_a \mathbf{V})_{ij}, & M_{ij} = 0. \end{cases} \tag{3}$$

We are now ready to elaborate on the details of the three disentangled representation matrices.

## 4.2 TREND REPRESENTATION MATRIX

Trend representation matrix $\mathbf{V}_t$ characterizes the intrinsic trend of time series. The evolution patterns dominated by $\mathbf{V}_t$ are supposed to change gradually and smoothly in the absence of accidents

or extreme events such as holidays (we will consider these external interventions and exogenous influences in $\mathbf{V}_b$). Based on this, we impose a smoothness constraint on $\mathbf{V}_t$ as

$$f_t(\mathbf{V}_t) = \sum_{j=2}^{T} \|\mathbf{v}_t^j - \mathbf{v}_t^{j-1}\|^2, \tag{4}$$

where $\mathbf{v}_t^j$ is the $j$-th column of $\mathbf{V}_t$. Equation 4 encourages close representations of two adjacent time steps in latent space, which will result in a smooth change in data space. We only impose constraints on two consecutive time steps to account for short-term patterns here whereas long-term patterns are explained by $\mathbf{V}_s$. This is in contrast with TRMF (Yu et al., 2016) which uses one temporal matrix to account for both short-term and long-term patterns by imposing regression constraint.

## 4.3 SEASONALITY REPRESENTATION MATRIX

Real-world time series often demonstrate seasonal patterns. For instance, traffic flow exhibits strong daily and weekly seasonality due to regularity of human activities, such as commuting patterns. Solar power production presents clear periodic characteristics caused by climate seasonality and meteorological conditions. Motivated by this and Fourier analysis in Section A.1.2, we propose to model the seasonality of time series by parameterizing representation matrix $\mathbf{V}_s$ with Fourier basis.

$\mathbf{V}_s$ is a matrix with size $D \times T$, we represent each row with a superposition of $2K$ sinusoidal waves ($K \ll T$). More formally, let $\mathbf{A}, \mathbf{B} \in \mathbb{R}^{D \times K}$ be two learnable coefficient matrices, and $\phi_{\sin}$, $\phi_{\cos} \in \mathbb{R}^{T \times K}$ be the corresponding Fourier basis matrices, which are defined as

$$\mathbf{A} = \left[ \begin{array}{cccc} | & | & & | \\ \mathbf{a}_1 & \mathbf{a}_2 & \dots & \mathbf{a}_K \\ | & | & & | \end{array} \right], \quad \phi_{\sin} = \left[ \begin{array}{cccc} | & | & & | \\ \sin(1\omega\mathbf{t}) & \sin(2\omega\mathbf{t}) & \dots & \sin(K\omega\mathbf{t}) \\ | & | & & | \end{array} \right], \tag{5}$$

$$\mathbf{B} = \left[ \begin{array}{cccc} | & | & & | \\ \mathbf{b}_1 & \mathbf{b}_2 & \dots & \mathbf{b}_K \\ | & | & & | \end{array} \right], \quad \phi_{\cos} = \left[ \begin{array}{cccc} | & | & & | \\ \cos(1\omega\mathbf{t}) & \cos(2\omega\mathbf{t}) & \dots & \cos(K\omega\mathbf{t}) \\ | & | & & | \end{array} \right], \tag{6}$$

where $\mathbf{t} = [1, \dots, T]^\top$. For one specific time-series with period $P$, $\omega$ is calculated as $2\pi/P$.

Then the seasonality representation matrix $\mathbf{V}_s$ can be defined as

$$\mathbf{V}_s = \mathbf{A}\phi_{\sin}^\top + \mathbf{B}\phi_{\cos}^\top. \tag{7}$$

In other words, $\mathbf{V}_s$ is spanned by Fourier basis $\phi_{\sin}$ and $\phi_{\cos}$. In particular, the $d$-th row of $\mathbf{V}_s$, denoted by $(\mathbf{v}_s^d)^\top$, has the form

$$(\mathbf{v}_s^d)^\top = \sum_{k=1}^{K} \mathbf{A}_{d,k} \sin(k\omega\mathbf{t})^\top + \sum_{k=1}^{K} \mathbf{B}_{d,k} \cos(k\omega\mathbf{t})^\top, \tag{8}$$

which is a truncated Fourier series with coefficients $\mathbf{A}_{d,k}, \mathbf{B}_{d,k}$. This elaborate design of $\mathbf{V}_s$ provides meaningful periodic inductive bias, which enables our model to capture seasonal patterns more accurately and effectively, by learning coefficient matrices $\mathbf{A}, \mathbf{B}$ from data.

## 4.4 BIAS TEMPORAL REPRESENTATION MATRIX

The representation matrices $\mathbf{V}_t$, $\mathbf{V}_s$ presented so far jointly determine the dynamics driven by endogenous factors of multivariate time series. However, there are also various external factors (e.g., holidays, weekdays/weekends) that could affect real-world time series. These external factors usually occur at specific time points and yield local variations within short time period, thus they are independent of endogenous dynamics and cannot be captured by $\mathbf{V}_t$ and $\mathbf{V}_s$. And also, these factors impact nearly equally on all channels. To account for these local variations, inspired by the idea of user and item bias explored in collaborative filtering (Lü et al., 2012), we propose to learn another bias representation matrix $\mathbf{V}_b \in \mathbb{R}^{D_b \times T}$, where the representation of a specific time step is shared by all channels. In addition, we impose an autoregressive constraint on $\mathbf{V}_b$ in temporal dimension since the impact caused by local variation usually lasts for a short duration. Let $\mathbf{v}_t$ be the $t$-th column of $\mathbf{V}_b$ and $L$ be the maximum time lag indicating duration, we define the constraint function as follows,

$$f_b(\mathbf{V}_b) = \sum_{t=L+1}^{T} \left\| \mathbf{v}_b^t - \sum_{l=1}^{L} \mathbf{W}_l \mathbf{v}_b^{t-l} \right\|^2, \tag{9}$$

where $\mathcal{W} = \{\mathbf{W}_l \in \mathbb{R}^{D_b \times D_b} \mid l = 1 \ldots L\}$ is a group of learnable parameters. In our setting, $D_b$ and $L$ are small numbers, thus group $\mathcal{W}$ only incurs very few extra parameters.

## 4.5 Adaptive Weight for Trend and Seasonality

In Section 4.1, we use the additive form $\mathbf{V}_t + \mathbf{V}_s$ to characterize the influence of endogenous impacting factors. This implicitly assumes that trend and seasonality components contribute equally to endogenous dynamics. But in practice the importance of trend and seasonality can vary drastically across data sources, which is illustrated in Figure 2. To address this, we adopt a learnable parameter $\alpha \in (0, 1)$ to adaptively adjust the weight for these two components, which leads to a weighted additive form $\alpha \mathbf{V}_t + (1 - \alpha)\mathbf{V}_s$. The temporal matrix $\mathbf{V}$ in Equation 2 then becomes:

$$\mathbf{V} = \left[ \begin{array}{c} \alpha \mathbf{V}_t + (1 - \alpha)\mathbf{V}_s \\ \mathbf{V}_b \end{array} \right]. \tag{10}$$

## 5 Experiments

In this section, we evaluate the performance of TIDER by comparing with existing multivariate time series imputation methods, in terms of imputation accuracy and scalability. We also show the explanability of TIDER with several case studies. Hyperparameter sensitivity experiments are included as well to show that TIDER performs steadily under different hyperparameter settings. The code of TIDER is available at `https://github.com/liuwj2000/TIDER`

### 5.1 Experimental Setup

**Baseline Methods** We compare our model with popular baselines used in the literature and recently-proposed methods, including statistical models (S333 impleMean, KNN, MICE), MF-based methods (MF, MF+L2, SoftImpute, TRMF), and deep learning approaches (BRITS, GAIN, NAOMI, SingleRes, SAITS, CSDI). Details and settings of these baselines can be found in Section A.3. In addition, we also include TIDER (no W), a variant of TIDER without the learnable parameter $\alpha$, to verify the effectiveness of the adaptive weight introduced in Section 4.5.

**Datasets** We use three typical real-world datasets. Guangzhou is a small one where all methods can fit in. Solar-energy shares along time-span while Westminster is one with large N. These datasets represent three different types of multivariate time series (small, large T, large N), and these three types can cover most of the multivariate time series. For more details, please refer to Section A.2.

**Evaluation Metrics** We adopt RMSE, MAE, and MAPE to evaluate the imputation accuracy of all compared methods. Details of these three metrics can be found in Section A.4.

**Training Setup** We randomly remove a subset of entries from $\mathbf{X}$ as validation and test datasets separately. Let $r$ be the missing rate variable, the ratio of training/validation/test is $(0.9 - r)/0.1/r$. For each model we run experiments 7 rounds on every dataset and report the imputed results averaged over these 7 runs. All experiments are conducted on a Linux workstation with a 32GB Tesla V100 GPU. For more detailed hyperparameter settings, please refer to Table 7.

### 5.2 Imputation Accuracy Comparison

Table 1 and Table 2 show the imputation accuracy of all methods on three datasets with different missing rate $r$. OOM indicates out of memory. The meaning of asterisk and improvement are illustrated in Section A.3. Our proposed TIDER achieves the best performance in most cases in terms of the three metrics. The superiority of TIDER is much more significant in Solar-energy, the dataset with large $T$. In addition, we obtain similar improvements on other long time series. We report experiments on another two datasets in Section A.5. In addition, we observe that the imputation accuracy of most methods drops as missing rate $r$ increases, which is as expected since fewer patterns are available.

Among all baseline methods, CSDI has the best performance on most datasets, and deep-learning methods (BRITS, GAIN, SAITS, CSDI) usually perform better than other baselines. However, TIDER outperforms them on each dataset. Especially on the solar-energy dataset, where time span $T$ is $52,560$ and many deep learning methods run into OOM on a 32GB-memory GPU, TIDER is

Table 1: Imputation accuracy of different methods with missing rate $r = 0.2$

| Method ($r = 0.2$) | Guangzhou | | | Solar-energy | | | Westminster | | |
|---|---|---|---|---|---|---|---|---|---|
| | RMSE | MAE | MAPE | RMSE | MAE | MAPE | RMSE | MAE | MAPE |
| SimpleMean | 10.24 | 8.023 | 0.396 | 3.352 | 2.433 | 4.527 | 5.233 | 4.247 | 0.410 |
| KNN | 6.540 | 4.731 | 0.242 | 2.701 | 1.748 | 2.324 | 2.249 | 1.611 | 0.147 |
| MF | 5.642 | 4.301 | 0.188 | 3.769 | 2.447 | 4.243 | 2.275 | 1.641 | 0.137 |
| MF+L2 | 5.400 | 4.074 | 0.177 | 2.761 | 1.918 | 2.760 | 2.109 | 1.523 | 0.134 |
| SoftImpute | 5.876 | 4.465 | 0.217 | 2.692 | 1.903 | 3.051 | 2.347 | 1.645 | 0.141 |
| MICE | 5.097 | 3.859 | 0.161 | 2.647 | 1.763 | 2.372 | 2.638 | 2.014 | 0.169 |
| TRMF | 4.563 | 3.407 | 0.148 | 3.212 | 2.010 | 2.937 | 1.903 | 1.370 | 0.119 |
| BRITS | 4.416 | 3.003 | 0.139 | 2.617 | 1.861 | 2.677 | 2.154* | 1.488* | 0.136* |
| GAIN | 4.976 | 3.451 | 0.154 | 2.803 | 1.917 | 2.501 | 1.947 | 1.403 | 0.121 |
| SAITS | 4.407 | 3.025 | 0.140 | OOM | OOM | OOM | 1.893 | 1.366 | 0.119 |
| CSDI | 4.301 | **2.991** | 0.135 | OOM | OOM | OOM | 1.886 | 1.361 | 0.116 |
| NAOMI | 5.173 | 4.013 | 0.167 | OOM | OOM | OOM | OOM | OOM | OOM |
| SingleRes | 4.997 | 3.979 | 0.172 | OOM | OOM | OOM | OOM | OOM | OOM |
| TIDER (no W) | 4.431 | 3.229 | 0.142 | 1.872 | 0.893 | 2.522 | 1.981 | 1.426 | 0.127 |
| TIDER | **4.168** | 3.098 | **0.132** | **1.676** | **0.874** | **2.227** | **1.867** | **1.354** | **0.115** |
| Improvement (%) | 3.092 | / | 2.222 | 35.96 | 50.00 | 4.174 | 1.007 | 0.514 | 0.862 |

Table 2: Imputation accuracy of different methods with missing rate $r = 0.4$

| Method ($r = 0.4$) | Guangzhou | | | Solar-energy | | | Westminster | | |
|---|---|---|---|---|---|---|---|---|---|
| | RMSE | MAE | MAPE | RMSE | MAE | MAPE | RMSE | MAE | MAPE |
| SimpleMean | 9.949 | 7.901 | 0.384 | 3.259 | 2.412 | 4.889 | 5.243 | 4.259 | 0.407 |
| KNN | 6.712 | 4.955 | 0.252 | 2.724 | 1.835 | 2.874 | 2.391 | 1.714 | 0.159 |
| MF | 7.671 | 5.844 | 0.246 | 4.111 | 2.887 | 5.851 | 2.476 | 1.807 | 0.153 |
| MF+L2 | 7.324 | 5.553 | 0.235 | 2.617 | 1.80 | 2.977 | 2.157 | 1.593 | 0.134 |
| SoftImpute | 5.811 | 4.465 | 0.209 | 2.536 | 1.849 | 3.311 | 2.333 | 1.637 | 0.145 |
| MICE | 7.119 | 5.417 | 0.235 | 2.756 | 1.823 | 3.517 | 2.755 | 2.113 | 0.175 |
| TRMF | 6.119 | 4.617 | 0.198 | 3.407 | 2.199 | 3.586 | 2.014 | 1.453 | 0.126 |
| BRITS | 4.874 | 3.335 | 0.158 | 2.842 | 1.985 | 3.146 | 2.180* | 1.527* | 0.138* |
| GAIN | 5.550 | 3.671 | 0.192 | 2.639 | 1.883 | 3.117 | 2.337 | 1.711 | 0.145 |
| SAITS | 4.839 | 3.391 | 0.159 | OOM | OOM | OOM | 1.998 | 1.453 | 0.129 |
| CSDI | 4.813 | **3.202** | 0.157 | OOM | OOM | OOM | 1.982 | 1.437 | 0.124 |
| NAOMI | 5.986 | 4.543 | 0.222 | OOM | OOM | OOM | OOM | OOM | OOM |
| SingleRes | 6.051 | 4.705 | 0.252 | OOM | OOM | OOM | OOM | OOM | OOM |
| TIDER (no W) | **4.708** | 3.469 | 0.155 | 1.697 | 0.878 | 3.152 | 2.013 | 1.466 | 0.125 |
| TIDER | 4.764 | 3.527 | **0.152** | **1.679** | **0.838** | **2.735** | **1.959** | **1.422** | **0.121** |
| Improvement (%) | 2.182 | / | 3.185 | 33.79 | 54.03 | 4.836 | 1.160 | 1.055 | 2.419 |

still applicable and achieves the best performance by large margins. This could be attributed to our model's ability to model multiple factors in representation space in a disentangled way.

We observe that MF-based models can all work for long time series. The difference between these models lies in constraints on **U** and **V**. Compared with other MF-based baselines, TRMF outperforms others in most cases, since it intuitively captures auto-regressive dynamics. However, since TRMF employs merely an entangled representation, it cannot model complex dynamics well. TIDER (no W) and TIDER utilize disentangled features like trend, seasonality and local bias to rebuild patterns for time-series, which exploit explanatory temporal factors, thus achieving better imputation result. Furthermore, TIDER performs better than TIDER (no W). This is consistent with our previous analysis that trend and seasonality components might not contribute equally to global dynamics.

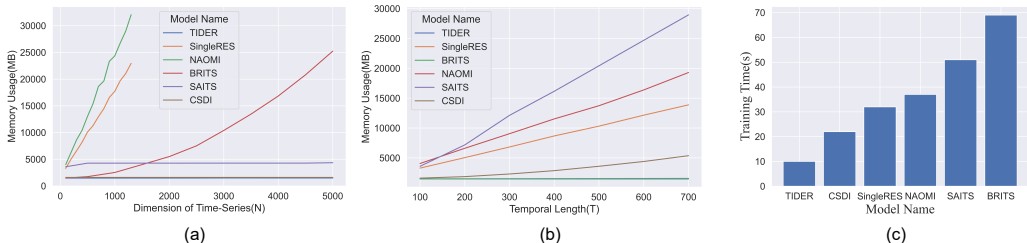

Figure 3: Scalability test on Westminster dataset: (a) Memory usage of different methods varies over $N$ from 0 to 5000 with fixed $T = 100$; (b) Memory usage of different methods varies over $T$ from 100 to 700 with fixed $N = 100$; (c) The average running time taken by different methods for every 100 epochs when processing a $100 \times 100$ matrix.

Table 3: Ablation Analysis of TIDER on Guangzhou dataset.

| Name | RMSE | MAE | MAPE |
|---|---|---|---|
| $\mathbf{V}_s + \mathbf{V}_b$ | 7.241 (7.497) | 5.388 (5.459) | 0.272 (0.294) |
| $\mathbf{V}_t + \mathbf{V}_b$ | 5.680 (6.392) | 4.311 (4.899) | 0.190 (0.214) |
| $\mathbf{V}_s + \mathbf{V}_t$ | 5.297 (5.999) | 4.014 (4.520) | 0.170 (0.201) |
| TIDER | **4.168 (4.764)** | **3.098 (3.527)** | **0.132 (0.152)** |

## 5.3 SCALABILITY ANALYSIS

We study the scalability of different methods in memory usage and training time. In particular, we compare our method with state-of-the-art models, BRITS, NAOMI, SingleRes, SAITS, and CSDI. Figure 3-(a) and Figure 3-(b) present the memory footprints of different methods against channel number $N$ and time span $T$. It can be seen from Figure 3-(a) that memory usage of NAOMI and SingleRes grow rapidly with very steep slopes, and that of BRITS also grows quickly whereas TIDER needs much less memory. Similarly, Figure 3-(b) shows that the memory usage of NAOMI, SingleRes, and SAITS also grows fast as $T$ increases. Again, TIDER needs the least amount of memory. There is also a gap between the curves of CSDI (BRITS) and TIDER in Figure 3, which is overwhelmed by the magnitude of NAOMI and SingleRes. We present these gaps in Section A.9. Lastly, Figure 3-(c) demonstrates the running time of different methods for 100 epochs when processing a $100 \times 100$ matrix. TIDER runs much faster than RNNs-based methods. Notably, TIDER outperforms BRITS by almost an order of magnitude. This is also true for the total time taken by entire training process. Furthermore, the space complexity of BRITS is $O(N(N+T))$(Vaswani et al., 2017) while that of TIDER is $O(N+T)$. The complexity of NAOMI was not established and is tricky to analyze due to its complex divide-and-conquer strategy, but its complicated procedure yields more intermediate variables and thus results in more memory consumption, as empirically validated by our experiments. In conclusion, this experiment demonstrates the scalability of our model.

## 5.4 ABLATION ANALYSIS

To verify the effectiveness of our proposed disentangled representation matrices, we conduct an ablation study on Guangzhou dataset by removing one of the trend, seasonality, and bias representation matrices while leaving the rest of the model unchanged. The ablative results with missing rate $r = 0.2\,(0.4)$ are presented in Table 3. We find that the model performance drops no matter which representation is removed, which validates that all of our proposed disentangled representations play an important role in imputation and jointly enhance the performance of the final model.

## 5.5 CASE STUDY OF DISENTANGLEMENT

This section is to demonstrate that the learned representations achieve disentanglement and offer explainability. We first visualize a time-series from Guangzhou dataset and its learnt patterns, namely, trend patterns $\mathbf{u}^\top \mathbf{V}_t$, seasonality patterns $\mathbf{u}^\top \mathbf{V}_s$, and bias patterns $\mathbf{1}^\top \mathbf{V}_b$, where $\mathbf{u}^\top$ is the time-series representation (a row of $\mathbf{U}$). Figure 4 shows that the learned representations generate semantically meaningful patterns. Figure 4-(b) shows a gradually changing curve that corresponds to trend

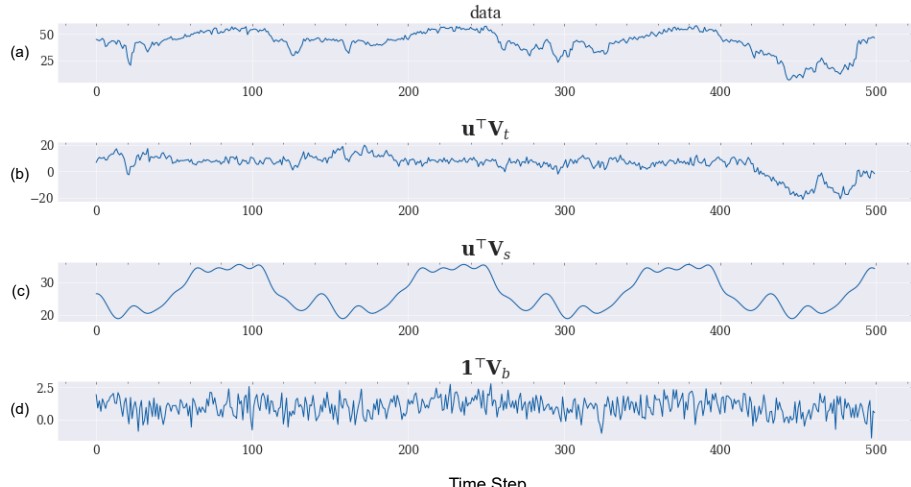

Figure 4: Case study of disentanglement. (a) is the original time-series, whereas (b),(c) and (d) represent the generated intrinsic trend, seasonality, and local variation within a period of time.

patterns, whereas Figure 4-(c) exhibits clear periodicity, and Figure 4-(d) manifests more idiosyncratic behaviors. Each of the displayed patterns captures information from a particular semantic perspective, which verifies that the learned representations $\mathbf{V}_t$, $\mathbf{V}_s$, $\mathbf{V}_b$ exhibit independence and disentanglement.

In order to further show TIDER's interpretability, we conduct experiments on synthetic time-series where one pair of time-series are composed of the same trend but different seasonality, while another pair of time-series are composed of the same seasonality but a different trend. We visualize the ground truth trend and seasonality patterns, together with the learned ones. Their results are depicted in Figure 5 and Figure 6. We can clearly find that the learned disentangled components are very similar to the ground truth, while the disentangled patterns that are supposed to be close to each other indeed look similar (similar slopes on the trend components, together with similar amplitudes and periods on the seasonality components). This further supports TIDER's explainability of disentanglement.

### 5.6 HYPERPARAMETER SENSITIVITY

We analyze the impact of four key hyperparameters ($D$, $K$, $D_d$, and $P$) on performance of TIDER, and present the results in Figure 7, 8, 9, and 10, respectively. We observe that TIDER is relatively stable under different hyperparameter settings. Even for hyperparameter $P$ (period), TIDER exhibits consistent imputation performance, although interpretability may be compromised. Therefore, if accurate imputation is the primary objective, extensive hyperparameter tuning may not be necessary, as TIDER is robust to variations in hyperparameters. However, when both accuracy and interpretability are important, determining the appropriate time series period is critical. This can often be achieved through prior knowledge or other periodic detection models (Fan et al., 2022; Wang et al., 2022).

### 6 CONCLUSION

In this paper, we propose a scalable multivariate time series imputation method, TIDER, with multiple novel inductive biases under the framework of low-rank matrix factorization. In contrast to existing imputation approaches, TIDER adopts semantically meaningful disentangled representations to account for multiple factors of a time series. In particular, it enables capturing periodicity with a novel Fourier basis-based representation and allows to identify local time variation with a time bias representation. TIDER's superiority is verified by experimental results. Moreover, it scales well to long time series where the existing methods struggle or even cannot fit. In the future, it is interesting to investigate how to use the meaningful disentangled representations for forecasting tasks and how to design constraints on $\mathbf{U}$ when channel information is available. Furthermore, since now we use hyperparameter tuning techniques to decide $P$ in $\mathbf{V}_s$, how to obtain $P$ adaptively in a data-driven way is also worth further investigation.

# 7 REPRODUCIBILITY STATEMENT

We provide an open-source implementation of our proposed model, TIDER, at `https://github.com/liuwj2000/TIDER`. Hyperparameter setting is shown in Section A.8. Users can download the code and run TIDER easily.

# 8 ACKNOWLEDGEMENT

This study is supported under the RIE2020 Industry Alignment Fund – Industry Collaboration Projects (IAF-ICP) Funding Initiative, as well as cash and in-kind contribution from Singapore Telecommunications Limited (Singtel), through Singtel Cognitive and Artificial Intelligence Lab for Enterprises (SCALE@NTU). This research/project is also supported by the National Research Foundation, Singapore under its AI Singapore Programme (AISG Award No: AISG2-TC-2021-001).

Xiucheng Li is supported in part by the National Natural Science Foundation of China under Grant No. 62206074, Shenzhen College Stability Support Plan under Grant No. GXWD20220811173233001. The main part of his job is done when he is in Nanyang Technological University.

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

## A  APPENDIX

### A.1  PRELIMINARY

#### A.1.1  LOW-RANK MATRIX FACTORIZATION

Given a matrix $\mathbf{X} \in \mathbb{R}^{N \times T}$ with rank $k \ll \min\{N, T\}$, the low-rank matrix factorization aims to factorize $\mathbf{X}$ as the product of two low-rank matrices $\mathbf{U} \in \mathbb{R}^{N \times k}$ and $\mathbf{V} \in \mathbb{R}^{k \times T}$,

$$\mathbf{X} = \mathbf{UV}. \tag{11}$$

This low-rank matrix factorization can be extended to the circumstance where the true rank of $\mathbf{X}$ is larger than $k$. In such a case, we use a metric $\ell$ to measure the discrepancy between $\mathbf{X}$ and the approximated product $\mathbf{UV}$ as

$$\ell(\mathbf{X}, \mathbf{UV}) = \|\mathbf{X} - \mathbf{UV}\|^2, \tag{12}$$

where $\| \cdot \|$ denotes the Frobenius norm of a matrix. This low-rank approximation is fairly reasonable in practice, since many real-world structural observation matrices are low rank. This has been widely adopted and extensively studied in recommender systems (Takács et al., 2008; Chen et al., 2020). In the context of multivariate time series data, we propose to adopt low-rank matrix factorization for two reasons: 1) univariate time series channels (the rows of $\mathbf{X}$) are usually highly correlated; 2) observations at different time steps (the columns of $\mathbf{X}$) exhibit strong dependencies.

Table 4: Summary of dataset statistics.

| Name | Number of channels ($N$) | Number of time steps ($T$) |
|---|---|---|
| Guangzhou[1] | 214 | 500 |
| Solar-Energy[2] | 137 | 52,560 |
| Westminster[3] | 7,489 | 744 |

### A.1.2 FOURIER SERIES

Fourier analysis(Stein & Shakarchi, 2011) states that any real-valued periodic function $f(x)$ can be represented by an infinite series of sinusoidal functions, known as the Fourier series. These sinusoidal functions have their own coefficients and distinct frequencies. More formally, the Fourier series for a function $f(x)$ with its period $P$ can be described as

$$f(x) = \sum_{n=0}^{\infty} a_n \cos(n\omega x) + \sum_{n=0}^{\infty} b_n \sin(n\omega x),  \tag{13}$$

where $\omega = 2\pi/P$, and $a_n, b_n \in \mathbb{R}$ are the corresponding coefficients. Since $\sin(n\omega x)$ and $\cos(n\omega x)$ functions are orthogonal to each other and are able to span over the entire functional space, they are also called Fourier basis.

### A.2 DATASET DETAILS

- **Guangzhou Traffic Data.** This dataset(Chen et al., 2018) contains traffic speed of 214 anonymous urban road segments for 5 days with a 10-minute sampling rate in Guangzhou, China. It results in a $214 \times 500$ multivariate time series matrix.

- **Solar-Energy Production Data.** This dataset consists of solar power production records of 137 PV plants in Alabama, USA sampled every 10 minutes. It results in a $137 \times 52,560$ data matrix.

- **Westminster Uber Movement Data.** This dataset contains hourly averaged speed of road segments in Westminster, in Jan 2020, released by Uber. Dimension of its data matrix is $7,489 \times 744$.

The dataset statistics are summarized in Table 4.

### A.3 DETAILS OF BASELINE MODELS

The details of baseline methods are briefly summarized as follows. For SimpleMean, KNN, and SoftImpute, we use their implementation provided by the package fancyimpute whereas for BRITS, GAIN, SAITS, CSDI, NAOMI and SingleRes, we use the source codes released by their authors.

- **SimpleMean**(Acuna & Rodriguez, 2004)[4] It imputes missing entries with mean values of corresponding columns.

- **KNN**(Batista et al., 2002)[5] It first finds $k$ nearest rows with the highest similarity score to the target row, and then uses the weighted sum of these $k$ rows for imputation.

- **SoftImpute**(Mazumder et al., 2010)[6]. It is a matrix completion approach based on iterative soft thresholding of Singular Value Decomposition (SVD).

- **MICE**(Azur et al., 2011)[7]. It is the abbreviation for Multivariate Imputation by Chained Equations, a widely-used R package for imputation.

- **MF**(Takács et al., 2008). It applies low-rank matrix factorization without any constraint on the latent factors.

---

[4]`https://github.com/iskandr/fancyimpute`
[5]`https://github.com/iskandr/fancyimpute`
[6]`https://github.com/iskandr/fancyimpute`
[7]`https://github.com/amices/mice`

- MF+L2(Takács et al., 2008). It applies low-rank matrix factorization with L2 regularization on $\mathbf{U}$ and $\mathbf{V}$.
- TRMF(Yu et al., 2016). It applies low-rank matrix factorization with an autoregressive constraint imposed on temporal matrix $\mathbf{V}$.
- BRITS(Cao et al., 2018)[8]. It is a time series imputation method based on Bidirectional Recurrent Neural Nets and a time decay mechanism.
- GAIN(Yoon et al., 2018)[9]. It employs Generative Adversarial Nets (GAN) for imputation.
- SAITS(Du et al., 2022)[10]. It imputes missing values based on self-attention mechanism.
- CSDI(Tashiro et al., 2021)[11]. It utilizes score-based diffusion models to explore correlations between observed values and for imputation.
- NAOMI(Liu et al., 2019)[12]. It combines Bidirectional Recurrent Neural Nets with adversarial training to offer non-autoregressive style imputation. The divide-and-conquer strategy is adopted.
- SingleRes(Liu et al., 2019)[13]. It is the autoregressive counterpart of NAOMI.

In our experiments, we find that BRITS with the suggested batch size and RNN hidden size will result in OOM (Out of Memory) on the Westminster dataset. Thus we reduce the batch size to 1 and the dimension of RNN to 10 such that it could just fit the 32 GB GPU on that dataset, we use an asterisk symbol to indicate its results in Table 1 and Table 2, in which the Improvement is calculated as follows,

$$\text{Improvement} = \frac{\text{best\_baseline} - \text{TIDER}^*}{\text{best\_baseline}} \times 100\% \tag{14}$$

where best_baseline represents the best performance among all the compared baseline models, $\text{TIDER}^*$ stands for the better one between TIDER and TIDER (no W).

### A.4 DETAILS OF METRICS

Root Mean Square Error (RMSE), Mean Absolute Error (MAE), and Mean Absolute Percentage Error (MAPE) are adopted to evaluate the imputation accuracy of all compared methods. These three metrics are defined as

$$\text{RMSE} = \sqrt{\frac{\sum_{ij \in \Omega}(\mathbf{X}_{ij} - \hat{\mathbf{X}}_{ij})^2}{|\Omega|}}, \text{MAE} = \frac{\sum_{ij \in \Omega}|\mathbf{X}_{ij} - \hat{\mathbf{X}}_{ij}|}{|\Omega|}, \text{MAPE} = \sum_{ij \in \Omega} \frac{|\mathbf{X}_{ij} - \hat{\mathbf{X}}_{ij}|}{|\Omega| \cdot |\mathbf{X}_{ij}|}. \tag{15}$$

where $\mathbf{X}_{ij}$ denotes the ground-truth values, $\hat{\mathbf{X}}_{ij}$ is the imputed values, and $\Omega$ is the index set of missing entries to be evaluated.

### A.5 ADDITIONAL EXPERIMENTS FOR IMPUTATION ACCURACY COMPARISON ON LONG TIME SERIES

As Table 1 and Table 2 shows, TIDER outperforms the baseline models by large margins on long time-series dataset Solar-Energy. In order to further verify its performance stability in imputing long multivariate time-series data, we conduct additional experiments on another two long time-series:

- HouseHold Power dataset [14]. This dataset contains every minute's household electric consumption measurements gathered in a house located in Sceaux from December 2006 to November 2010. The size of this long time-series matrix is $7 \times 2075259$.

---

[8]https://github.com/caow13/BRITS
[9]https://github.com/jsyoon0823/GAIN
[10]https://github.com/WenjieDu/SAITS
[11]https://github.com/ermongroup/CSDI
[12]https://github.com/felixykliu/NAOMI
[13]https://github.com/felixykliu/NAOMI
[14]https://archive.ics.uci.edu/ml/datasets/Individual+household+electric+power+consumption

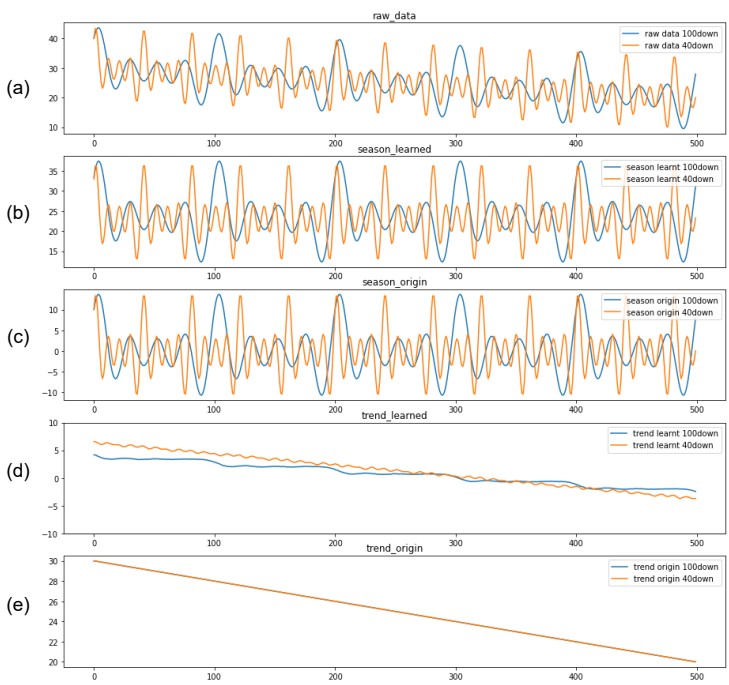

Figure 5: Disentanglement validation on synthetic dataset. Time series are composed of the same trend but different seasonalities. (a) shows the raw time series. (b) and (d) show the trend and seasonality components TIDER has learned. (c) and (e) depict the ground trend and seasonality components which compose the raw time series.

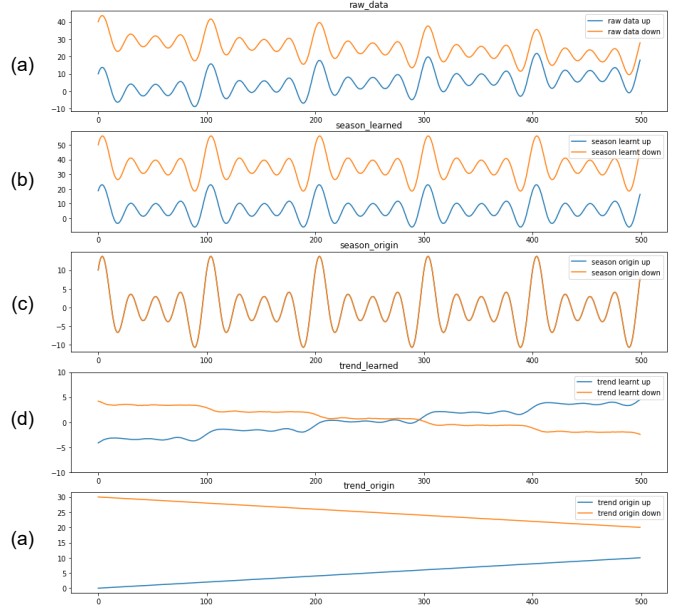

Figure 6: Disentanglement validation on synthetic dataset. Time series are composed of different trends but the same seasonality. (a) shows the raw time series. (b) and (d) show the trend and seasonality components TIDER has learned. (c) and (e) depict the ground trend and seasonality components which compose the raw time series.

Table 5: Imputation accuracy of different methods with missing rate $r = 0.2$, for long time series.

| Method | HouseHold Power | | | Beijing Climate | | |
|---|---|---|---|---|---|---|
| ($r = 0.2$) | RMSE | MAE | MAPE | RMSE | MAE | MAPE |
| SimpleMean | 100.6 | 70.07 | 10.69 | 418.5 | 295.9 | 6.521 |
| KNN | 91.57 | 38.55 | 4.313 | 406.9 | 184.5 | 1.141 |
| MF | 87.88 | 33.75 | 0.885 | 404.3 | 177.4 | 1.005 |
| MF+L2 | 82.62 | 29.79 | 0.791 | 393.5 | 170.4 | 1.033 |
| SoftImpute | 85.61 | 33.75 | 0.651 | 386.5 | 172.9 | 1.043 |
| MICE | 79.21 | 31.27 | 0.642 | 292.2 | 111.9 | 0.827 |
| TRMF | 89.14 | 35.39 | 0.685 | 312.7 | 110.1 | 0.811 |
| BRITS | 66.98 | 25.61 | 0.335 | 101.9 | 66.42 | 0.611 |
| GAIN | 70.04 | 25.92 | 0.498 | 125.4 | 62.37 | 0.607 |
| SAITS | OOM | OOM | OOM | OOM | OOM | OOM |
| CSDI | OOM | OOM | OOM | OOM | OOM | OOM |
| NAOMI | OOM | OOM | OOM | OOM | OOM | OOM |
| SingleRes | OOM | OOM | OOM | OOM | OOM | OOM |
| TIDER (no W) | 56.36 | 20.20 | 0.351 | 185.8 | 99.78 | 0.754 |
| TIDER | **53.95** | **18.29** | **0.284** | **60.65** | **32.77** | **0.532** |
| Improvement (%) | 24.15 | 28.58 | 15.22 | 40.48 | 47.45 | 12.28 |

Table 6: Imputation accuracy of different methods with missing rate $r = 0.4$, for long time series.

| Method | HouseHold Power | | | Beijing Climate | | |
|---|---|---|---|---|---|---|
| ($r = 0.4$) | RMSE | MAE | MAPE | RMSE | MAE | MAPE |
| SimpleMean | 105.2 | 68.17 | 9.064 | 459.8 | 313.8 | 6.454 |
| KNN | 96.07 | 42.68 | 5.362 | 397.8 | 179.4 | 1.053 |
| MF | 90.70 | 36.72 | 1.494 | 388.4 | 164.9 | 1.015 |
| MF+L2 | 86.71 | 34.06 | 1.262 | 384.7 | 162.8 | 1.013 |
| SoftImpute | 85.72 | 33.61 | 0.698 | 385.1 | 168.0 | 0.912 |
| MICE | 83.46 | 32.48 | 0.672 | 281.1 | 111.3 | 0.820 |
| TRMF | 101.5 | 57.36 | 0.985 | 312.5 | 110.5 | 0.807 |
| BRITS | 69.82 | 27.82 | 0.398 | 163.0 | 75.69 | 0.676 |
| GAIN | 74.47 | 26.93 | 0.512 | 177.3 | 73.79 | 0.658 |
| SAITS | OOM | OOM | OOM | OOM | OOM | OOM |
| CSDI | OOM | OOM | OOM | OOM | OOM | OOM |
| NAOMI | OOM | OOM | OOM | OOM | OOM | OOM |
| SingleRes | OOM | OOM | OOM | OOM | OOM | OOM |
| TIDER (no W) | 64.44 | 25.53 | 0.391 | 221.4 | 112.0 | 0.872 |
| TIDER | 59.07 | 20.01 | 0.356 | 80.87 | 55.85 | 0.581 |
| Improvement (%) | 15.40 | 25.70 | 10.55 | 50.39 | 24.31 | 11.70 |

- Beijing Climate dataset [15]. This dataset records the PM2.5 concentration, dew point, temperature, pressure, combined wind direction, cumulative wind speed, cumulative hours of snow, and cumulative hours of rain in Beijing, from Jan 1st, 2010 to Dec 31st, 2014. The size of this long time-series matrix is $7 \times 43824$.

The imputation accuracy of all models on these two long time series is shown in Table 5 and Table 6. Similar to the observation made in Section 5.2, many deep-learning-based models that perform well in short time-series dataset Guangzhou (eg, CSDI, SAITS) run into out of memory on these two datasets, while our proposed method can easily handle them and achieve significantly better performance. These results further support the scalability of our model, as well as its effectiveness in imputing long time-series datasets.

---

[15] https://archive.ics.uci.edu/ml/datasets/Beijing+PM2.5+Data

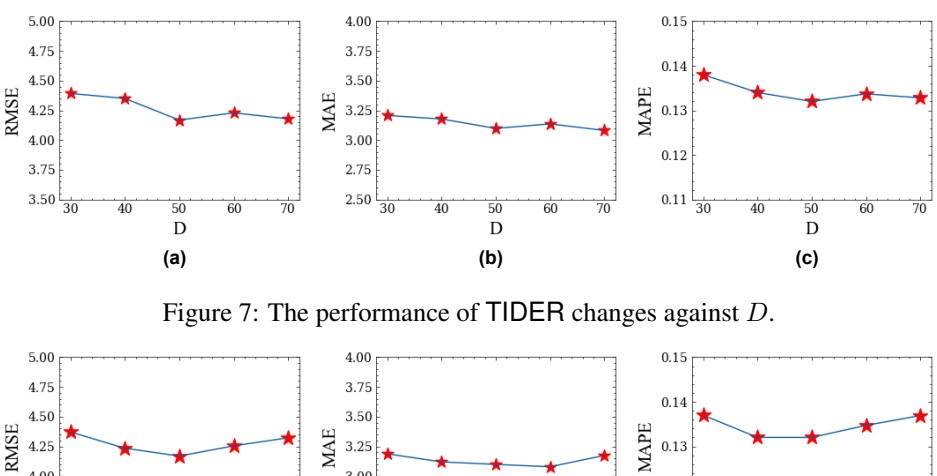

Figure 7: The performance of TIDER changes against $D$.

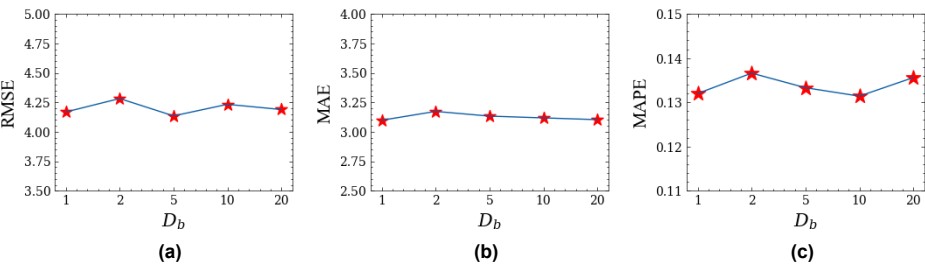

Figure 8: The performance of TIDER changes against $K$.

## A.6 DISENTANGLEMENT VALIDATION ON SYNTHETIC DATASET

In order to further show TIDER's interpretability, we conduct experiments on synthetic time-series where one pair of time-series are composed of the same trend but different seasonality, while another pair of time-series are composed of the same seasonality but a different trend. We visualize the ground truth trend and seasonality patterns, together with the learned ones. Their results are depicted in Figure 5 and Figure 6. We can clearly find that the learned disentangled components are very similar to the ground truth, while the disentangled patterns that are supposed to be close to each other indeed look similar (similar slopes on the trend components, together with similar amplitudes and periods on the seasonality components). This further supports TIDER's explainability of disentanglement.

## A.7 HYPERPARAMETER SENSITIVITY

In this section, we study the performance change of TIDER under different hyperparameter settings. Figure 7 shows TIDER's hyperparameter sensitivity under different $D$. Figure 8 draws the performance of TIDER changing against $K$, the number of sinusoidal waves in $\mathbf{V}_s$. Figure 7 depicts the accuracy curves of TIDER varying over different $D_d$. Figure 10 is used to study the performance stability of TIDER under different periods in $\mathbf{V}_s$. It can be seen that TIDER is rather stable under different hyperparameter settings. Therefore, if our target is merely for accurate imputation, we might not need to pay much effort or spend much time on hyperparameter tuning, since TIDER is stable among different hyperparameter settings. If we target both accuracy and interpretability, then the only hyperparameter we need to figure out is the period of time series. This often can be acquired by our prior knowledge or by other periodic detection models Wen et al. (2021).

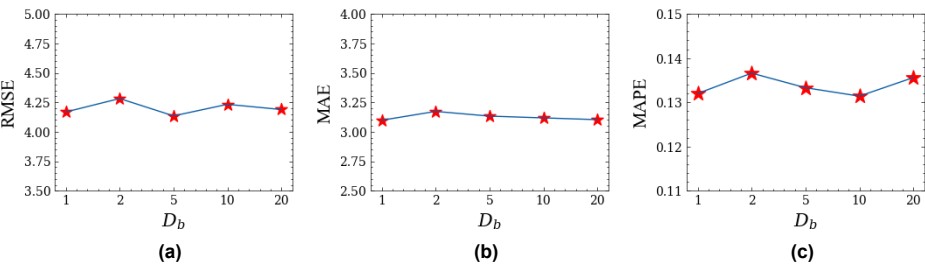

Figure 9: The performance of TIDER changes against $D_d$.

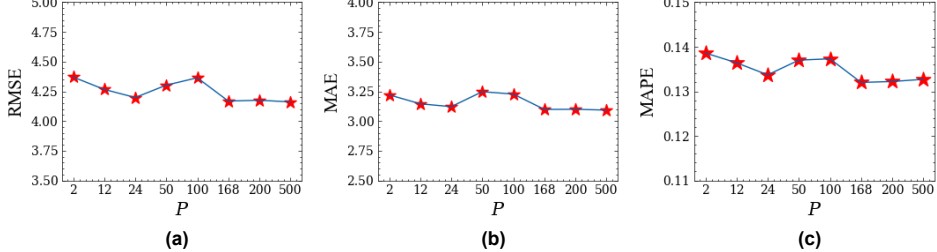

Figure 10: The performance of TIDER changes against $P$.

Table 7: Hyperparameters of TIDER for the three datasets.

| Name ($r = 0.2$) | Guangzhou | Solar-Energy | Westminster |
|---|---|---|---|
| batch size | 32 | 32 | 512 |
| epoch | 500 | 500 | 500 |
| $D$ | 50 | 50 | 150 |
| $D_b$ | 2 | 2 | 2 |
| $\eta_1$ | 0.01 | 0.01 | 0.01 |
| $\eta_2$ | 0.01 | 0.01 | 0.01 |
| $\lambda_t$ | 0.1 | 0.01 | 0.01 |
| $\lambda_b$ | 0.2 | 0.1 | 0.1 |
| $K$ | 20 | 2 | 15 |
| $\omega$ | $\frac{\pi}{84}$ | $\frac{\pi}{84}$ | $\frac{\pi}{12}$ |
| **Name ($r = 0.4$)** | **Guangzhou** | **Solar-Energy** | **Westminster** |
| batch size | 32 | 32 | 512 |
| epoch | 500 | 500 | 500 |
| $D$ | 50 | 50 | 150 |
| $D_b$ | 2 | 2 | 2 |
| $\eta_1$ | 0.01 | 0.01 | 0.01 |
| $\eta_2$ | 0.01 | 0.01 | 0.01 |
| $\lambda_t$ | 0.1 | 0.01 | 0.01 |
| $\lambda_b$ | 0.2 | 0.1 | 0.1 |
| $K$ | 20 | 2 | 15 |
| $\omega$ | $\frac{\pi}{84}$ | $\frac{\pi}{84}$ | $\frac{\pi}{12}$ |

## A.8 HYPERPARAMETER SETTING

We implement TIDER using Python 3.6 and Pytorch 1.9, and optimize the model parameters using Adam (Kingma & Ba, 2014) with a learning rate $1e-3$. We use gird search to select the optimal hyperparameters ($D$, $K$, $D_d$, $P$) on the validation datasets. However, as we have observed in Section 5.6 and Section A.8, the imputation performance of TIDER is relatively steady among different hyperparameter settings. Thus other hyperparameter sets might also offer desirable results.

The general principle and guidelines on parameter setting are as follows:

- $\omega$: This hyperparameter is involved in the seasonality representation matrix. $\omega$ is in close relation with the time-series period $P$. At present, $P$ is chosen by jointly using our prior knowledge of different time series and hyperparameter tuning techniques. We first construct a candidate list of $P$ based on our knowledge of these time series and select the best one according to performance on validation datasets. We set $P$ for Guangzhou, Solar-Energy, and Westminster datasets as 168, 168, and 24 respectively. In other words, the optimal $\omega$ for these datasets are $\frac{\pi}{84}$, $\frac{\pi}{84}$ and $\frac{\pi}{12}$.

- $D$: Dimension for matrix $\mathbf{U}$ and $\mathbf{V}_t$ is also an important hyperparameter. Small $D$ will lack the ability to learn enough information while large $D$ is prone to overfitting.

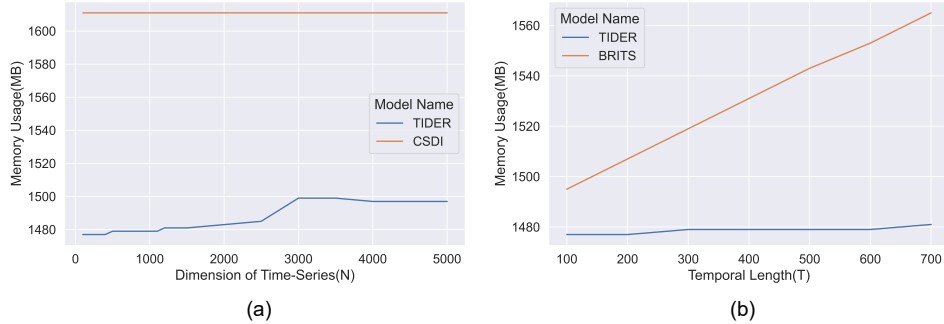

Figure 11: Supplementary scalability test on Westminster dataset. (a) Memory usage between TIDER and CSDI varies over $N$ from 0 to 5000 with fixed $T = 100$; (b) Memory usage between TIDER and BRITS varies over $T$ from 100 to 700 with fixed $N = 100$.

- $\lambda_t, \lambda_b$: These are the corresponding weights for the constraint functions of the trend representation matrix and bias representation matrix, respectively. We tune them by making a choice from a scale set $\{0.01, 0.1, 0.2, 0.5, 1.0\}$.

By following this principle, the optimal hyperparameters we obtained are listed in Table 7.

### A.9 SUPPLEMENTAL SCALABILITY ANALYSIS

In Section 5.3, there is a little gap between memory usage of CSDI and TIDER in Figure 3(a), and little gap between memory usage of BRITS and TIDER in Figure 3(b). In Figure 3-(a) (resp. Figure 3-(b)) the memory usage gap between CSDI (resp. BRITS) and TIDER is overwhelmed by the large magnitude of other methods. Thus, we zoom in to give a comparison in this section. As depicted in Figure 11, there are still large memory usage gaps between TIDER and them, which demonstrates the scalability superiority of our proposed TIDER.

