# OpenReview forum: "Multivariate Time-series Imputation with Disentangled Temporal Representations"
_ICLR.cc/2023/Conference — ICLR 2023 poster_

### Official Review · Reviewer_tDGy · 2022-10-17

**Confidence:** 3
**Clarity, Quality, Novelty And Reproducibility:** Good
**Correctness:** 2
**Technical Novelty And Significance:** 3
**Empirical Novelty And Significance:** 3
**Recommendation:** 6

**Strength And Weaknesses:**

Pros
1. Using matrix factorization for time series imputation is novel.
2. This method is scalable to long time series
3. Figure 4 shows how this method is more interpretable than deep learning based methods. The figure clearly shows  trend, seasonality, and local variation of the dataset

Cons
1. I am not sure the assumption of  trend, seasonality, and local variation hold for all the datasets. There may be some datasets that don't have trends or seasonality and some datasets contain very unpredictable pattern
2. I am not sure how this work can impute missing data by factorization. Could you explain more?
3. The sentence "but also scales well to large datasets on which existing deep learning based methods struggle." in abstract is misleading. What the author actually want to say is those deep learning methods is not scalable to long time series rather than large datasets.

**Summary Of The Paper:**

This paper propose a disentangled matrix factorization method for time series imputation. This paper achieves better accuracy and better interpretability compared to several baselines.

**Summary Of The Review:**

This paper is novel overall and the experiments are promising.

---

> ### Author Response · Authors · 2022-11-17
> **Response to Reviewer tDGy (1/2)**
>
> Q1
>
> >"I am not sure the assumption of trend, seasonality, and local variation hold for all the datasets. There may be some datasets that don't have trends or seasonality and some datasets contain very unpredictable pattern"
>
> A2
>
> Thank you for your comment. Treating time-series as a combination of trend, seasonality (and bias) is very **common** in the literature on time-series e.g., [1, 2, 3]. This can cover a **wide spectrum** of multivariate time series data, and many models are developed under this assumption and prove to be effective. Similarly, TIDER is already suitable for a wide spectrum of time series data. It would be difficult for one technique to handle all datasets. What’s more, as [4] proves, it is fundamentally **impossible** to learn disentangled representations **without** inductive bias on models and data.
>
> [1]  Jinwen Qiu, S Rao Jammalamadaka, and Ning Ning. Multivariate bayesian structural time series model. J. Mach. Learn. Res., 19(1):2744–2776, 2018.
>
> [2]  Wen Q, Gao J, Song X, et al. RobustSTL: A robust seasonal-trend decomposition algorithm for long time series[C]//Proceedings of the AAAI Conference on Artificial Intelligence. 2019, 33(01): 5409-5416.
>
> [3]  Woo G, Liu C, Sahoo D, et al. CoST: Contrastive Learning of Disentangled Seasonal-Trend Representations for Time Series Forecasting[C]//International Conference on Learning Representations. 2021.
>
> [4] Locatello, Francesco, et al. "Challenging common assumptions in the unsupervised learning of disentangled representations." international conference on machine learning. PMLR, 2019.
>
> Q2
>
> > "I am not sure how this work can impute missing data by factorization. Could you explain more?"
>
> A2
>
> Thank you for your comment. In short, there are often hidden structures (instance correlation and temporal dynamics) in time series data  and the factorization can explore such structures to impute. The structures imply that different entries in the time series matrix are correlated, e.g., the observed entries and missing entries, and thus we can learn the corresponding representations (or factors) from the observed entries and then apply the learned representations into the missing ones to infer their values.
> More specifically, the explanation can be further divided into two parts: how to perform imputation by matrix factorization, and how our proposed disentangled representation makes imputation more accurate and more explainable.
>
> Firstly, multivariate time-series imputation intends to use the small fraction of data with observations to impute the missing elements. In general, if no constraints are placed on time series matrix, the missing elements can take any values. However, most of the time series matrices preserve certain structures, i.e., instance correlation and temporal dynamics. Both the observed entries and missing entries are dominated by such structures, as a result, it offers us the opportunity to learn the representations on the observed entries and apply them to impute the missing entries since such structure shares across entries.
>
> Secondly, to improve imputation accuracy, many models in recent years have added other regularization terms or other structures into matrix factorization to perform the imputation, and they have obtained good results to some extent.  However, as we have mentioned in our introduction and related work, these imputation approaches are all based on single entangled representations (hidden states) to model time series for imputation. These entangled representations would be compromised to account for multiple orthogonal factors of one time-series, thus leading to inferior performance and lack of explainability. Meanwhile, treating time-series as a combination of trend, seasonality (and bias) is very common and widely-used in time-series related works, and this can cover a wide spectrum of multivariate time series data [3, 4, 5]. Inspired by these, we propose to use disentangled representation matrices  to account for different factors of one time series, separately. These disentangled representations bring stronger inductive bias and more powerful expressiveness to model and impute time series.
>
> [1] M.Fazel. Matrix Rank Minimization with Applications. PhD thesis, Stanford University.
>
> [2] R.Meka. Guaranteed rank minimization via singular value projection. In Proceedings of Advances in Neural Information Processing Systems. 2009
>
> [3] Jinwen Qiu, S Rao Jammalamadaka, and Ning Ning. Multivariate bayesian structural time series model. J. Mach. Learn. Res., 19(1):2744–2776, 2018.
>
> [4]  Wen Q, Gao J, Song X, et al. RobustSTL: A robust seasonal-trend decomposition algorithm for long time series[C]//Proceedings of the AAAI Conference on Artificial Intelligence. 2019, 33(01): 5409-5416.
>
> [5]  Woo G, Liu C, Sahoo D, et al. CoST: Contrastive Learning of Disentangled Seasonal-Trend Representations for Time Series Forecasting[C]//International Conference on Learning Representations. 2021.

---

> > ### Comment · Reviewer_tDGy · 2022-12-06
> > **thanks for clarification**
> >
> > thanks for pointing out the references that also have the assumption of trend and seasonality and the clarification about imputation. My concerns are addressed and I prefer to keep my score.

---

> ### Author Response · Authors · 2022-11-17
> **Response to Reviewer tDGy (2/2)**
>
> Q3
>
> > "The sentence "but also scales well to large datasets on which existing deep learning based methods struggle." in abstract is misleading. What the author actually want to say is those deep learning methods is not scalable to long time series rather than large datasets."
>
>
> A3:
>
> We sincerely apologize for this misleading description. We have revised the corresponding parts in our article. Thank you for pointing it out.

---

> ### Author Response · Authors · 2022-12-04
> **Gentle Reminder**
>
> Dear Reviewer tDGy,
>
> We sincerely thank you again for your valuable comments and constructive suggestions. Since discussion stage 2 is approaching its end, we would appreciate it if you could please let us know whether our responses and the revised manuscript have addressed your concerns, and let us know if any issues remain.
>
> Thanks, Author of paper 751

---

### Official Review · Reviewer_3zzf · 2022-10-24

**Confidence:** 3
**Correctness:** 3
**Technical Novelty And Significance:** 3
**Empirical Novelty And Significance:** 3
**Recommendation:** 6

**Clarity, Quality, Novelty And Reproducibility:**

Clarity and Quality: Apart from redundancy issues discussed in above response the paper is clearly written.
Novelty: To the best of my knowledge this is indeed a novel contribution.
Reproducibility: Code is available and all hyperparameter choices are clearly listed in the paper.

**Strength And Weaknesses:**

Strengths:
- The paper is overall well written, well motivated, the problem and the solution both are very clearly articulated.
- This is a problem worth solving well as multivariate time series data are present in crucial applications and almost always require imputation. (Health monitoring is perhaps the most critical and obvious application of this)
- The experiments include comparisons with a wide variety of competing and baseline methods.
- The solution results in interpretable factors and is scalable.

Weakness:
- There is quite a bit of redundancy in the writing. e.g. the point that disentangled representations are better than entangled representations is made unnecessarily too many times. e.g. the first 6 lines of first paragraph in section 4.1 are not really needed in my opinion as that has already been said thrice earlier.

- I'd have loved to see more discussion of the interpretability/explainability aspect in the main paper, more than just a single case study in experiments section. If redundancy is removed there's plenty of room for discussion about this in the paper.

- My biggest concern is the objective function involves a lot of hyperparameters. And while the authors did a good job (from reproducibility perspective) of writing out all hyperparameter choices it is very much unclear how: 1) those choices can actually be made in practice (the current explanation in the appendix is too vague) and 2) how sensitive are the empirical results to hyperparameter choices. This second point is especially crucial since wrong choices can conceivably wipe out whatever improvement is gained from this method. I will be willing to reconsider my rating if this particular issue is resolved.

**Summary Of The Paper:**

The paper tackles the problem of imputing missing values in multivariate time series. As opposed to existing methods which depend on learning a single entangled representation to impute missing values, this paper proposes a disentangled matrix factorization based approach that aims to capture trend, seasonality and local bias as independent factors. This allows the proposed methods to be interpretable as well as scalable. The experiments claim to show improvement over existing methods.

**Summary Of The Review:**

The paper is good and can conceivably make it to ICLR. It is a worthwhile contribution but the extensive hyperparameter tuning involved here without clear guidance is a sticking point.

---

> ### Author Response · Authors · 2022-11-17
> **Response to Reviewer 3zzf**
>
> Q1
>
> >"There is quite a bit of redundancy in the writing. e.g. the point that disentangled representations are better than entangled representations is made unnecessarily too many times. e.g. the first 6 lines of first paragraph in section 4.1 are not really needed in my opinion as that has already been said thrice earlier."
>
> A1
>
> Thank you for your valuable suggestion. We have revised our manuscript and removed the redundant parts.
>
> Q2
>
> >"I'd have loved to see more discussion of the interpretability/explainability aspect in the main paper, more than just a single case study in experiments section. If redundancy is removed there's plenty of room for discussion about this in the paper."
>
> A2
>
> Thank you for your advice. We have rewritten the related parts. Please refer to the revised Section 5.5 for the updated interpretability discussion parts.
>
> Q3
>
> >"My biggest concern is the objective function involves a lot of hyperparameters. And while the authors did a good job (from reproducibility perspective) of writing out all hyperparameter choices it is very much unclear how: 1) those choices can actually be made in practice (the current explanation in the appendix is too vague)and 2) how sensitive are the empirical results to hyperparameter choices. This second point is especially crucial since wrong choices can conceivably wipe out whatever improvement is gained from this method. I will be willing to reconsider my rating if this particular issue is resolved."
>
> A3
>
> Thank you for your thoughtful comments. We first answer the second question, and then the first question.
>
> Firstly, following your advice, we conducted experiments on hyperparameter sensitivity. The resulting figures and their corresponding analysis are available in Section 5.6 and Section A.7. From these plots we can draw a conclusion that TIDER is relatively **stable** under different hyperparameter settings. Even for hyperparameter P (period), TIDER's imputation performance is also quite **stable** under different values (though the interpretability might be damaged under undesirable Ps). Therefore, if our target is merely for accurate imputation, we might not need to pay much effort or spend much time on hyperparameter tuning, since TIDER is stable among different hyperparameter settings. If we target both accuracy and interpretability, then the only hyperparameter we need to figure out is the period of time series. This often can be acquired by our prior knowledge or by other periodic detection models [1]. In addition, as we state in the conclusion, it is also worth further exploration on how to obtain the period of one time-series adaptively in a data-driven way, and we will leave it as our future work.
>
> Secondly, for hyperparameter choice, it can be done by applying grid search to select the best set of hyperparameters on the validation datasets. However, as we have observed in hyperparameter sensitivity experiments, TIDER’s imputation performance is quite stable among different hyperparameter settings. Thus, it will not take too much effort and time for the practitioners to tune the hyperparameters to obtain desirable results. We have revised our description regarding hyperparameter choices to make it clearer.
>
> [1] Wen Q, He K, Sun L, et al. RobustPeriod: Robust time-frequency mining for multiple periodicity detection[C]//Proceedings of the 2021 International Conference on Management of Data, SIGMOD 2021: 2328-2337.

---

> > ### Comment · Reviewer_3zzf · 2022-11-17
> > **Response to Authors**
> >
> > Thank you for your detailed response and making appropriate changes to the the paper. I will take this response and changes into account when providing final recommendation.

---

### Official Review · Reviewer_CdFR · 2022-10-25

**Confidence:** 4
**Correctness:** 4
**Technical Novelty And Significance:** 2
**Empirical Novelty And Significance:** 2
**Recommendation:** 5

**Clarity, Quality, Novelty And Reproducibility:**

This is a high-quality paper. Methods and experimental results are clearly explained, making it easy to reproduce the work. The method is simple but works very well. The explorations in the paper and the comprehensive experiments could contribute to this rarely-studied field.

**Strength And Weaknesses:**

Strengths:
1) This work is significant for the multivariate time-series imputation community. It focuses on two rarely studied directions -- explaining the temporal representations and using the disentangled temporal representations to improve imputation performance, which is still a challenging task in both classical and deep learning based methods.
2) The paper is well written. Experiments are sufficient and comprehensive and can prove the reliability as well as the explainability of the proposed method.

Weaknesses:
1)  The proposed method is based on matrix decomposition and may not be novel enough for ICLR. Also, disentangled factors are pre-defined and hard coded in the model -- Trend, seasonality, and bias factors. In reality, however, the hidden temporal factors of time-series data may be unknown, might be more complex, and should be learned as well. The paper may have limitations if given high dimensional temporal data or if multivariate time-series have more complex inter-channel correlations and variations.
2) The training is specific for individual time-series data. The structure of latent space is not regularized over dataset to guarantee generalizable disentanglement.

**Summary Of The Paper:**

The paper studies the multivariate time-series imputation problem. Existing methods all rely on entangled representations. In contrast, this paper contributes a novel methods that explicitly models 3 disentangled temporal representations, making the imputation process more reliable and offers interpretability.

**Summary Of The Review:**

After evaluating the novelty of method, the writing quality, and experiments, I think this paper is marginally below the acceptance threshold.

---

> ### Author Response · Authors · 2022-11-17
> **Response to Reviewer CdFR (1/2)**
>
> Q1
>
> > "The proposed method is based on matrix decomposition and may not be novel enough for ICLR."
>
> A1
>
> Thank you for your comments. We would like to clarify the originality and novelty of our proposed TIDER as follows.
>
> * **The disentanglement provides effectiveness and interpretability**. Firstly, as we mentioned in our introduction and related work, the previous time-series imputation approaches are all based on single entangled representations (hidden states) to model time series for imputation. These entangled representations would be compromised to account for multiple orthogonal factors of one time-series, thus leading to inferior performance and lack of explainability. In contrast, our model is the **first** imputation model to use disentangled representations to account for different factors of one time series, separately. These disentangled representations bring stronger inductive bias and more powerful expressiveness to model and impute time series. In particular, our proposed TIDER is the **first** imputation model to introduce learnable Fourier series-based representation to intuitively capture periodic patterns of time series. Imputation accuracy experiments (Table 1, 2, 5, and 6) show that our proposed TIDER substantially outperforms all the other baselines, especially in long time series. Ablation study (Table 3) shows that each disentanglement plays an important role in improving the imputation accuracy. The disentanglement validation experiments (Figure 4, 5, 6) also demonstrate the interpretability of TIDER. These theoretical analyses and empirical results all substantiate that TIDER is more than an incremental work.
>
> * **Careful design of the combination enjoys scalability**. Secondly, our combination of matrix factorization and time-series decomposition is **nontrivial** and is dedicatedly designed. Rather than doing time-series decomposition in data space (which is the general approach adopted by the time-series decomposition models [1, 2]), TIDER conducts its disentanglement (decomposition) in representation space. The approach is new for time series imputation. The benefits are twofold: i) this can decouple the channel-wise interactions, thus matrix U only preserves the channel-specific features whereas the V matrices (V_t, V_s, and V_b) merely preserve temporal-specific patterns; ii) this provides much more scalability, enabling our model to better accommodate extremely long time series.
>
> * **TIDER represents a new idea and sheds light on a new direction**. Lastly, to the best of our knowledge, TIDER is the first model that introduces disentanglement into time-series imputation. TIDER could give the time-series imputation community a new direction: we can use disentangled representations rather than entangled representations for more accurate and more explainable imputation.
>
> References:
>
> [1] Abdollahi H. A novel hybrid model for forecasting crude oil price based on time series decomposition[J]. Applied energy, 2020, 267: 115035.
>
> [2] Wen Q, Gao J, Song X, et al. RobustSTL: A robust seasonal-trend decomposition algorithm for long time series[C]//Proceedings of the AAAI Conference on Artificial Intelligence. 2019, 33(01): 5409-5416.
>
> Q2
>
> > “Also, disentangled factors are pre-defined and hard coded in the model -- Trend, seasonality, and bias factors. In reality, however, the hidden temporal factors of time-series data may be unknown, might be more complex, and should be learned as well.”
>
> A2
>
> Thank you for your comment. Treating time-series as a combination of trend, seasonality (and bias) is very **common** in the literature on time-series e.g., [1, 2, 3]. This can cover a **wide spectrum** of multivariate time series data, and many models are developed under this assumption and prove to be effective. Similarly, TIDER is already suitable for a wide spectrum of time series data. It would be difficult for one technique to handle all the hidden temporal factors.  What’s more, as [4] proves, it is fundamentally **impossible** to learn disentangled representations **without** inductive bias on models and data.
>
>  [1]  Jinwen Qiu, S Rao Jammalamadaka, and Ning Ning. Multivariate bayesian structural time series model. J. Mach. Learn. Res., 19(1):2744–2776, 2018.
>
> [2]  Wen Q, Gao J, Song X, et al. RobustSTL: A robust seasonal-trend decomposition algorithm for long time series[C]//Proceedings of the AAAI Conference on Artificial Intelligence. 2019, 33(01): 5409-5416.
>
> [3]  Woo G, Liu C, Sahoo D, et al. CoST: Contrastive Learning of Disentangled Seasonal-Trend Representations for Time Series Forecasting[C]//International Conference on Learning Representations. 2021.
>
> [4] Locatello, Francesco, et al. "Challenging common assumptions in the unsupervised learning of disentangled representations." international conference on machine learning. PMLR, 2019.

---

> > ### Comment · Reviewer_CdFR · 2022-12-06
> > **Response to Authors**
> >
> > Thank you for your response and the extra experiments to support the claims in the paper. Most of my questions for this paper are answered and I have no more questions. However, although this paper has its specific significance, I still consider the novelty is not enough for ICLR. "TIDER is the first model that introduces disentanglement into time-series imputation." As far as I know, this is not true. For example, the authors mention this work [3] that also studies disentanglement and considers the same latent factors. The forecasting problem is very similar to imputation. Therefore, I am not willing to change my decision.

---

> > > ### Author Response · Authors · 2022-12-08
> > > **Additional Response**
> > >
> > > Dear Reviewer CdFR,
> > >
> > > Thank you for your valuable feedback. We admit that we are not the first to apply disentanglement to time series analysis. The point we want to make here is that disentanglement is just a general idea, what matters is **how to realize** it by considering the structure of the specific problems. In the forecasting problem [1] ([3] in our previous response), the authors achieve seasonality by performing FFT and rFFT on the latent representations, which could be **computationally expensive**. In contrast, we accomplish the seasonality by parameterizing the seasonality representations with Fourier basis and integrating it with the Matrix Factorization framework, which is able to guarantee **scalability**. Moreover, the forecasting problem [1] only needs to consider the trend and seasonality factors, which is **not sufficient** in imputation problems since exogenous influence can cause local fluctuation. To handle this, we also design a bias representation matrix, which is necessary to ensure **accurate** imputation results as validated by our ablation experiments. Last but not least, the model in [1] is a **two-step** framework, which needs the contrastive learning approach, COST, to first learn disentangled representations, and another regression model to further conduct the forecasting task. This means that the disentangled representations and forecasting model are learned separately in [1].  In contrast, our proposed TIDER is an **end-to-end** model, it completes the imputation task together with the representation learning process.
> > >
> > > For imputation and prediction tasks, we would like to state that though they are similar in many aspects, there exist many other parts which **differ** between these two tasks. On the one hand, as we state in our introduction, many time-series forecasting models assume an intact time-series, thus missing values  will impair their performance or even make them inapplicable. On the other hand, in time-series imputation settings, data before and after are both available. While in time-series forecasting settings, only historical data are feasible. Thus, some time-series imputation models can not fit in forecasting settings either. Therefore, just as [2] states, **‘imputation is not prediction’**.
> > >
> > > In summary, given by our previous response and the two above additional reasons, we think our model is novel.
> > >
> > > Hoping our reply this time will give you a much clearer view. Please do not hesitate if any further questions exist.
> > >
> > > [1] Contrastive Learning of Disentangled Seasonal-Trend Representations for Time Series Forecasting. Woo, Gerald, et al. ICLR 2022.
> > >
> > > [2] Van Buuren S. Flexible imputation of missing data[M]. CRC press, 2018.

---

> ### Author Response · Authors · 2022-11-17
> **Response to Reviewer CdFR (2/2)**
>
> Q3
>
> > “The paper may have limitations if given high dimensional temporal data or if multivariate time-series have more complex inter-channel correlations and variations.”
>
> A3
>
> Thank you for your comment. We would like to clarify that we focus on 'multivariate uni-dimensional time series' (i.e., the dimension of each time-series instance is one), which is commonly used in related work [1, 2]. It is an interesting future work to impute multivariate multi-dimensional time series.
>
> For complex inter-channel correlations and variations, it's a good point to gain some insights from it for better imputation. As mentioned in our conclusion part, it is a promising research direction to design novel and effective constraints on channel feature matrix U for better imputation when the channel information is available. We will explore it more in our future work.
>
> [1]  Woo G, Liu C, Sahoo D, et al. CoST: Contrastive Learning of Disentangled Seasonal-Trend Representations for Time Series Forecasting[J]. arXiv preprint arXiv:2202.01575, 2022.
>
> [2]  Yu H F, Rao N, Dhillon I S. Temporal regularized matrix factorization for high-dimensional time series prediction[J]. Advances in neural information processing systems, 2016, 29.
>
> Q4
>
> >"The training is specific for individual time-series data. The structure of latent space is not regularized over dataset to guarantee generalizable disentanglement."
>
> A4
>
> Thank you for your comment. In order to address your concerns, we have conducted several additional experiments to show the hyperparameter sensitivity of TIDER. The figures and their corresponding analysis can be found in Section 5.6 and Section A.7. From these experiments we can find that the performance of TIDER is rather stable under different hyperparameters. Thus it is probably not a necessity to spend much effort to tune the hyperparameters of TIDER for a specific time-series data, and thus it is not necessary to regularize the structure of latent space over datasets.

---

> ### Author Response · Authors · 2022-12-04
> **Gentle Reminder**
>
> Dear Reviewer CdFR,
>
> We sincerely thank you again for your valuable comments and constructive suggestions. Since discussion stage 2 is approaching its end, we would appreciate it if you could please let us know whether our responses and the revised manuscript have addressed your concerns, and let us know if any issues remain.
>
> Thanks, Author of paper 751

---

### Official Review · Reviewer_iGko · 2022-10-25

**Confidence:** 3
**Correctness:** 4
**Technical Novelty And Significance:** 2
**Empirical Novelty And Significance:** 2
**Recommendation:** 5

**Clarity, Quality, Novelty And Reproducibility:**

I found the paper is clearly written and easy to understand. Although the proposed method seems to be interesting with a good motivation, it is a bit hard to say it’s novel.


**Strength And Weaknesses:**

The proposed model provides two benefits. First, cross-channel correlations can be decouples such that it enables to selectively optimize channel-wise model parameters, which scales well to large datasets. Second, the proposed method models complex temporal dynamics with explainability as time-related information can be isolated into the latent vector V.

The experimental results on imputation accuracy with missing rates show that the proposed method slightly performs better than its baselines. It is hard to say the difference in accuracy performance between TIDER and its baselines statistically significant.


**Summary Of The Paper:**

This paper propose a time-series imputation technique, called TIDER, which utilizes matrix factorization with disentangled temporal representations. The proposed method is designed to learn disentangled representations to explain seasonality and trend and expected to scale well to large datasets. The authors validate their models with pre-existing imputation methods (e.g., KNN, BRITS, CSDI) on three real-world datasets.

**Summary Of The Review:**

The authors provides a good description about their motivation and proposed method to solve the targeted problem. Although the suggested method is interesting, I have a concern with its originality as the suggested method seems to be a simple combination of low-rank matrix factorization with a notion of seasonability and trend in time-series dataset. Further, the experimental results are not quite convincing since the suggested method slightly outperforms its baselines on most datasets.

---

> ### Author Response · Authors · 2022-11-17
> **Response to Reviewer iGko (1/2)**
>
> Q1
> >"The experimental results on imputation accuracy with missing rates show that the proposed method slightly performs better than its baselines. It is hard to say the difference in accuracy performance between TIDER and its baselines statistically significant."
>
> >"Further, the experimental results are not quite convincing since the suggested method slightly outperforms its baselines on most datasets."
>
> A1
>
> Thank you for your feedback. We purposely select the three datasets used in our experiments to represent **three different types of multivariate time series**: Guangzhou dataset represents time series with small N and small T; Solar-Energy dataset shows the models' performance on long time series (with large T); Westminster dataset represents for the cases in which N is very large. **The three types of data can cover most of the multivariate time-series**. Our experimental results on these three typical datasets show that our proposed TIDER is **scalable** and **effective**. TIDER’s superiority is more significant on the second dataset (**solar-energy**), which represents a class of datasets, i.e., **long time-series**; However the baseline models like CSDI and SAITS cannot handle this kind of datasets, although they can perform well on the other two types of dataset. In contrast, TIDER outperforms the best baseline by **more than 50% in terms of MAE** and  **35% in terms of RMSE** on this long time-span dataset.
>
> Although we only report results on one long time-span dataset in the paper, we obtain similar improvements on other long time series data. To demonstrate this, we report **new comparison experiments** on two more datasets. The dataset description, imputation results, and the corresponding analysis are available in Section A.5. We also present the experimental results below for your easy reference. As we can see from the tables, TIDER achieves significantly better performance than the baselines, which demonstrates its superior effectiveness in handling long time series.
>
> Table R1-1 (Household power consumption dataset)
>
> |Model (r=0.2)|RMSE|MAE|MAPE|
> |:----: |:----: |:----: |:----: |
> |Simple_mean|100.6|70.07|10.69|
> |KNN|91.57|38.55|4.313|
> |MF|87.88|33.75|0.885|
> |MF+L2|82.62|29.79|0.791|
> |SoftImpute|85.61|33.75|0.651|
> |MICE|79.21|31.27|0.642|
> |TRMF|89.14|35.39|0.685|
> |BRITS|66.98|25.61|0.335|
> |GAIN|70.04|25.92|0.498|
> |SAITS|OOM|OOM|OOM|
> |CSDI|OOM|OOM|OOM|
> |NAOMI|OOM|OOM|OOM|
> |SingleRes|OOM|OOM|OOM|
> |TIDER(NO W)|56.36|20.20|0.351|
> |**TIDER**|**53.95**|**18.29**|**0.284**|
> |**Improvement (%)**|**24.15**|**28.58**|**15.22**|
>
> |Model (r=0.4)|RMSE|MAE|MAPE|
> |:----: |:----: |:----: |:----: |
> |Simple_mean|105.2|68.17|9.064|
> |KNN|96.07|42.68|5.362|
> |MF|90.70|36.72|1.494|
> |MF+L2|86.71|34.06|1.262|
> |SoftImpute|85.72|33.61|0.698|
> |MICE|83.46|32.48|0.672|
> |TRMF|101.5|57.36|0.985|
> |BRITS|69.82|27.82|0.398|
> |GAIN|74.47|26.93|0.512|
> |SAITS|OOM|OOM|OOM|
> |CSDI|OOM|OOM|OOM|
> |NAOMI|OOM|OOM|OOM|
> |SingleRes|OOM|OOM|OOM|
> |TIDER(NO W)|64.44|25.53|0.391|
> |**TIDER**|**59.07**|**20.01**|**0.356**|
> |**Improvement (%)**|**15.40**|**25.70**|**10.55**|
>
> Table R1-2 (Beijing pm2.5 dataset)
> |Model (r=0.2)|RMSE|MAE|MAPE|
> |:----: |:----: |:----: |:----: |
> |Simple_mean|418.5|295.9|6.521|
> |KNN|406.9|184.5|1.141|
> |MF|404.3|177.4|1.005|
> |MF+L2|393.5|170.4|1.033|
> |SoftImpute|386.5|172.9|1.043|
> |MICE|292.2|111.9|0.827|
> |TRMF|312.7|110.1|0.811|
> |BRITS|101.9|66.42|0.611|
> |GAIN|125.4|62.37|0.607|
> |SAITS|OOM|OOM|OOM|
> |CSDI|OOM|OOM|OOM|
> |NAOMI|OOM|OOM|OOM|
> |SingleRes|OOM|OOM|OOM|
> |TIDER(NO W)|185.8|99.78|0.754|
> |**TIDER**|**60.65**|**32.77**|**0.532**|
> |**Improvement (%)**|**40.48**|**47.45**|**12.28**|
>
> |Model (r=0.4)|RMSE|MAE|MAPE|
> |:----: |:----: |:----: |:----: |
> |Simple_mean|459.8|313.8|6.454|
> |KNN|397.8|179.4|1.053|
> |MF|388.4|164.9|1.015|
> |MF+L2|384.7|162.8|1.013|
> |SoftImpute|385.1|168.0|0.912|
> |MICE|281.1|111.3|0.820|
> |TRMF|312.5|110.5|0.807|
> |BRITS|163.0|75.69|0.676|
> |GAIN|177.3|73.79|0.658|
> |SAITS|OOM|OOM|OOM|
> |CSDI|OOM|OOM|OOM|
> |NAOMI|OOM|OOM|OOM|
> |SingleRes|OOM|OOM|OOM|
> |TIDER(NO W)|221.4|112.0|0.872|
> |**TIDER**|**80.87**|**55.85**|**0.581**|
> |**Improvement (%)**|**50.39**|**24.31**|**11.70**|

---

> ### Author Response · Authors · 2022-11-17
> **Response to Reviewer iGko (2/2)**
>
> Q2
>
> >"Although the suggested method is interesting, I have a concern with its originality as the suggested method seems to be a simple combination of low-rank matrix factorization with a notion of seasonability and trend in time-series dataset."
>
> >“ Although the proposed method seems to be interesting with a good motivation, it is a bit hard to say it’s novel.”
>
> A2
>
> Thank you for your comments. We would like to clarify that TIDER is **NOT** a trivial combination of existing techniques (low-rank matrix factorization and time-series decomposition). The originality and novelty of TIDER are highlighted as follows.
>
> * **The disentanglement provides effectiveness and interpretability**. Firstly, as we mentioned in our introduction and related work, the previous time-series imputation approaches are all based on single entangled representations (hidden states) to model time series for imputation. These entangled representations would be compromised to account for multiple orthogonal factors of one time-series, thus leading to inferior performance and lack of explainability. In contrast, our model is the **first** imputation model to use disentangled representations to account for different factors of one time series, separately. These disentangled representations bring stronger inductive bias and more powerful expressiveness to model and impute time series. In particular, our proposed TIDER is the **first** imputation model to introduce learnable Fourier series-based representation to intuitively capture periodic patterns of time series. Imputation accuracy experiments (Table 1, 2, 5, and 6) show that our proposed TIDER substantially outperforms all the other baselines, especially in long time series. Ablation study (Table 3) shows that each disentanglement plays an important role in improving the imputation accuracy. The disentanglement validation experiments (Figure 4, 5, 6) also demonstrate the interpretability of TIDER. These theoretical analyses and empirical results all substantiate that TIDER is more than an incremental work.
>
> * **Careful design of the combination enjoys scalability**. Secondly, our combination of matrix factorization and time-series decomposition is **nontrivial** and is dedicatedly designed. Rather than doing time-series decomposition in data space (which is the general approach adopted by the time-series decomposition models [1, 2]), TIDER conducts its disentanglement (decomposition) in representation space. The approach is new for time series imputation. The benefits are twofold: i) this can decouple the channel-wise interactions, thus matrix U only preserves the channel-specific features whereas the V matrices (V_t, V_s, and V_b) merely preserve temporal-specific patterns; ii) this provides much more scalability, enabling our model to better accommodate extremely long time series.
>
> * **TIDER represents a new idea and sheds light on a new direction**. Lastly, to the best of our knowledge, TIDER is the **first** model that introduces disentanglement into time-series imputation. TIDER could give the time-series imputation community a new direction: we can use disentangled representations rather than entangled representations for more accurate and more explainable imputation.
>
> References:
>
> [1]  Abdollahi H. A novel hybrid model for forecasting crude oil price based on time series decomposition[J]. Applied energy, 2020, 267: 115035.
>
> [2]  Wen Q, Gao J, Song X, et al. RobustSTL: A robust seasonal-trend decomposition algorithm for long time series[C]//Proceedings of the AAAI Conference on Artificial Intelligence. 2019, 33(01): 5409-5416.

---

> ### Author Response · Authors · 2022-12-04
> **Gentle Reminder**
>
> Dear Reviewer iGko,
>
> We sincerely thank you again for your valuable comments and constructive suggestions. Since discussion stage 2 is approaching its end, we would appreciate it if you could please let us know whether our responses and the revised manuscript have addressed your concerns, and let us know if any issues remain.
>
> Thanks, Author of paper 751

---

### Author Response · Authors · 2022-11-17
**General Response To All Reviewers**

We sincerely thank all the reviewers for their valuable comments, which have helped to improve the quality of our paper. We are delighted to see reviewers agree that our model is **novel** (Reviewer 3zzf, Reviewer tDGy), **interesting** (Reviewer iGko), **scalable** (Reviewer iGko, Reviewer 3zzf, Reviewer tDGy), **interpretable** (Reviewer iGko, Reviewer CdFR, Reviewer 3zzf, Reviewer tDGy), and **easy to reproduce** (Reviewer CdFR, Reviewer 3zzf). We are also happy that reviewers find our paper **well-written** (Reviewer iGko, Reviewer CdFR, Reviewer 3zzf) and **well-motivated** (Reviewer iGko, Reviewer 3zzf), our **experiments sufficient and comprehensive** (Reviewer CdFR ,Reviewer 3zzf, Reviewer tDGy), and our **work significant** (Reviewer CdFR).

The new experiments and major revisions are summarized as  follows:
* We add new comparison experiments on two long time-series datasets to demonstrate the superiority  of TIDER in imputing long time series, compared with baselines.
* We perform hyperparameter sensitivity experiments to test TIDER's stability under different hyperparameter settings.
* We remove redundant parts, and add more interpretability analysis and hyperparameter sensitivity experiments into the main paper.
* We rewrite the descriptions that may be misleading.
* We rewrite confusing statements and address the concerns raised by reviewers.
* We revised our paper to account for all changes, and the revisions are highlighted in blue color in the revised manuscript for your convenience.

We hope that our answers and revision of our paper will address your concerns.

---

### Decision · Program_Chairs · 2023-01-20

**Decision:**

Accept: poster

**Justification For Why Not Higher Score:**

Technical novelty is not high and more experiments are needed to understand when the method does well and when it doesn't and why.

**Justification For Why Not Lower Score:**

The paper makes good progress on an important problem with good improvements in imputation accuracy and scalability.

**Metareview: Summary, Strengths And Weaknesses:**

This paper considers the problem of multi-variate time-series imputation. It proposes a method referred to as TIDER by employing matrix factorization with disentangled temporal representations. The key idea is to learn disentangled representations of time-series to explain seasonality, trend, and local bias; and scale to large time-series. Experimental results on three real-world datasets show good improvements over the baseline method. Overall, the paper is well-written and easy to follow. Some reviewers suggested that the technical novelty is not very high.

Some comments and suggestions:
- The paper claims that this is the first time disentangled representations are used for time-series imputation. I found that the following NeurIPS-2020 paper which does the same for video imputation: https://arxiv.org/pdf/2006.13391.pdf If the paper is accepted, I recommend the authors' to correct their claim.
- It would have been nice to do some more ablations and stress testing of the proposed approach. For example, what is the improvement with learning representations for a subset of seasonality, trend, and local bias (TIDER uses all three). Some more analysis to understand where the method breaks and why. This will be useful to also validate the assumptions/domain knowledge baked into the overall solution.

This is a paper outside my area of expertise, but I think the paper makes good progress on an important problem. Therefore, I recommend accepting the paper and strongly encourage the authors' to revise the paper to reflect the reviewer-author discussion for the final submission.

**Note From Pc:**

if the above contains the word "oral" or "spotlight" please see: "oral" presentation means -> notable-top-5% and "spotlight" means -> notable-top-25%. As stated in our emails, we are disassociating presentation type from AC recommendations